# Neural crest induction requires SALL4-mediated BAF recruitment to lineage specific enhancers

Martina Demurtas[1], Samantha M. Barnada[2], Emma van Domselaar[1], Zoe H. Mitchell[1], Laura Deelen[1] and Marco Trizzino[1,*]

## ABSTRACT

Neural crest induction begins early during neural plate formation, requiring precise transcriptional control to activate lineage-specific enhancers. Here, we demonstrate that SALL4, a transcription factor associated with syndromes featuring craniofacial anomalies, plays a crucial role in early cranial neural crest (CNCC) specification. Using *SALL4*-het-KO human iPSCs to model clinical haploinsufficiency, we show that SALL4 directly recruits BAF to CNCC-lineage specific enhancers at the neuroectodermal stage, specifically when neural crest gene expression is induced at the neural plate border. Without functional SALL4, BAF is not loaded at chromatin, leaving CNCC enhancers inaccessible. Consequently, the cells cannot undergo proper CNCC induction and specification due to persistent enhancer repression, despite normal neuroectodermal and neural plate progression. Moreover, by performing SALL4 isoform-specific depletion, we demonstrate that SALL4A is the isoform essential for CNCC induction and specification, and that SALL4B cannot compensate for SALL4A loss in this developmental process.

In summary, our findings reveal SALL4 as essential regulator of BAF-dependent enhancer activation during early stages of neural crest development, providing molecular insights into SALL4-associated craniofacial anomalies.

KEY WORDS: SALL4, DPF2, SALL4A, BAF, Neural crest, Enhancers, Chromatin accessibility

## INTRODUCTION

Transcription factors and chromatin remodellers play crucial roles in cell fate determination (Gopinathan and Diekwisch, 2022; Islam et al., 2021; Kim and Shendure, 2019; Li, 2002; Voss and Hager, 2014; Zheng et al., 2021). Among these, the C2H2 zinc-finger SALL4 is involved in a wide range of developmental processes (Tatetsu et al., 2016). SALL4 is highly expressed in pluripotent cells and throughout embryonic development, gradually decreasing after birth to become undetectable in adult tissues (Xiong et al., 2015).

In humans, *SALL4* produces two main isoforms: SALL4A (1053 aa) and SALL4B (617 aa), which have been proposed to play distinct roles in embryonic stem cells (ESCs). Namely, Rao et al. (2010) suggested that SALL4B has function limited to pluripotency regulation in ESCs, while SALL4A is primarily active during development. Although *SALL4* is expressed as early as the two-cell stage, its role in maintaining pluripotency remains under debate. Some studies suggest that SALL4 cooperates with core pluripotency factors to sustain stem cell identity (Yang, 2018; Zhang et al., 2006). In contrast, others propose that SALL4 plays a more marginal role in pluripotency regulation (Miller et al., 2016; Wang et al., 2014; Yuri et al., 2009).

During early mouse development, *Sall4* is broadly expressed throughout the embryo; however, as gastrulation progresses, its expression becomes more posteriorly restricted, localising primarily to the developing tail bud and limb buds (Chen et al., 2022; Tahara et al., 2019). Here, it regulates cell fate commitment of neuromesodermal progenitors by suppressing neural genes in mesodermal cells while simultaneously enhancing mesodermal gene expression (Pappas et al., 2024). Additionally, *Sall4* remains expressed in mouse craniofacial regions until embryonic day (E)9.5, including the developing frontonasal process and the mandibular and maxillary arches (Tahara et al., 2019). These craniofacial structures originate from cranial neural crest cells (CNCCs), a subpopulation of the neural crest cells that gives rise to the cartilage and bones of the face and anterior skull, as well as cranial neurons and glia (Dash and Trainor, 2020; Milet and Monsoro-Burq, 2012; Pla and Monsoro-Burq, 2018; Roth et al., 2021; Szabó and Mayor, 2018). Consistent with this, individuals with *SALL4* haploinsufficiency exhibit a range of phenotypes, including distinctive facial features such as epicanthal folds, widely spaced eyes, and a depressed and flattened nasal bridge, along with cranial nerve-related conditions and underdevelopment of one side of the face (Kekunnaya and Negalur, 2017). Many of these distinctive features are classic indicators of impaired CNCC development. *SALL4*-associated phenotypes include Duane anomaly, a congenital strabismus resulting from abnormal oculomotor nerve innervation (Kekunnaya and Negalur, 2017), and sensorineural hearing loss (Terhal et al., 2006), but also upper limb anomalies (Duane-radial ray syndrome) and, in some cases, renal anomalies. Together, these findings suggest that SALL4 may play a crucial role in CNCC development, which has not yet been investigated.

Previous studies have also suggested that SALL4 interacts directly with chromatin remodelling complexes. For example, Sall4 is an interactor of the NuRD complex in mouse ESCs and during development (Bode et al., 2016; Liu et al., 2018b; Lu et al., 2009; Miller et al., 2016; Wang et al., 2023). Additionally, recent mass-spectrometry experiments conducted by our group in early stages of human iPSC-to-CNCC differentiation, and specifically at the neuroectodermal stage, have revealed that SALL4 interacts with the BAF complex at this developmental stage (Pagliaroli et al., 2021). Since this was the first report of a potential direct SALL4-BAF interaction, here we set out to investigate the scope of this interaction and determine the role of SALL4 during CNCC development.

[1]Department of Life Sciences, Imperial College London, London SW7 2AZ, UK.
[2]Department of Biochemistry and Molecular Biology, Thomas Jefferson University, Philadelphia, PA 19107, USA.

*Author for correspondence (m.trizzino@imperial.ac.uk)

M.D., 0009-0004-9506-1697; S.M.B., 0000-0003-4316-4008; Z.H.M., 0009-0007-4315-8720; L.D., 0009-0007-2295-2554; M.T., 0000-0002-1383-7200

## RESULTS

### SALL4 interacts with the BAF complex

CNCCs can be generated *in vitro* using human induced pluripotent stem cells (hiPSCs). This protocol is 2 weeks long, and it first goes through a neuroectoderm intermediate (Bajpai et al., 2010; Barnada et al., 2024; Mitchell et al., 2025 preprint; Pagliaroli et al., 2021; Prescott et al., 2015). In our previous work (Pagliaroli et al., 2021), mass spectrometry (MS) conducted at the early stages of the iPSC-to-CNCC differentiation (day 5, corresponding to neuroectoderm) revealed that SALL4 interacts with BAF at this developmental stage. Since this was the first report of a SALL4-BAF interaction, we performed a co-immunoprecipitation (co-IP) at day 5 to validate it, pulling-down ARID1B, which is the only ARID1 subunit active and incorporated in BAF at this stage (Pagliaroli et al., 2021), and blotting for SALL4. This co-IP confirmed that ARID1B interacts with both SALL4 isoforms at this time-point (Fig. 1A).

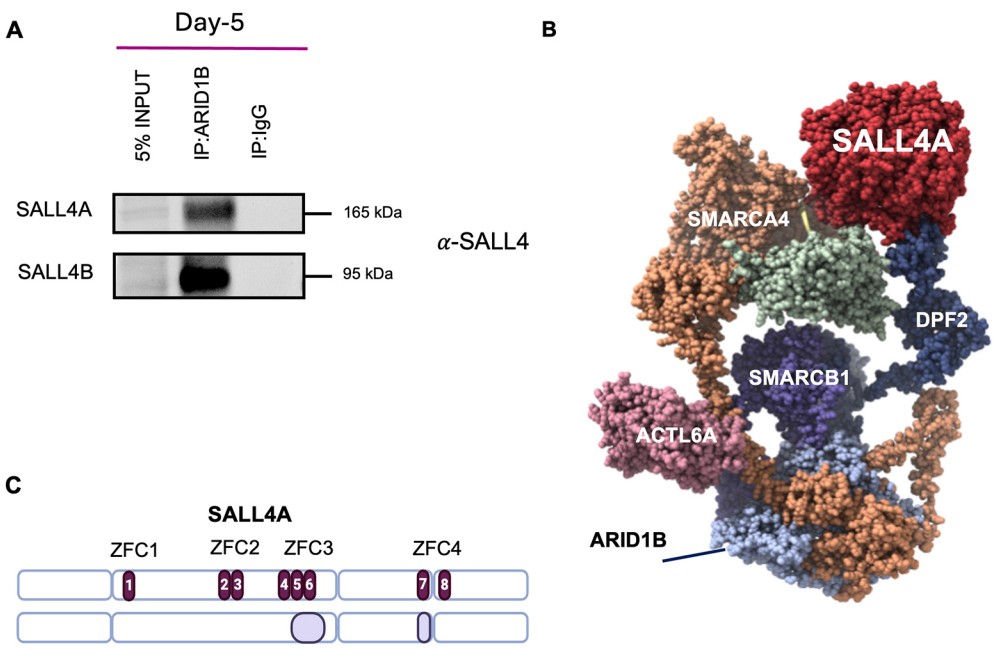

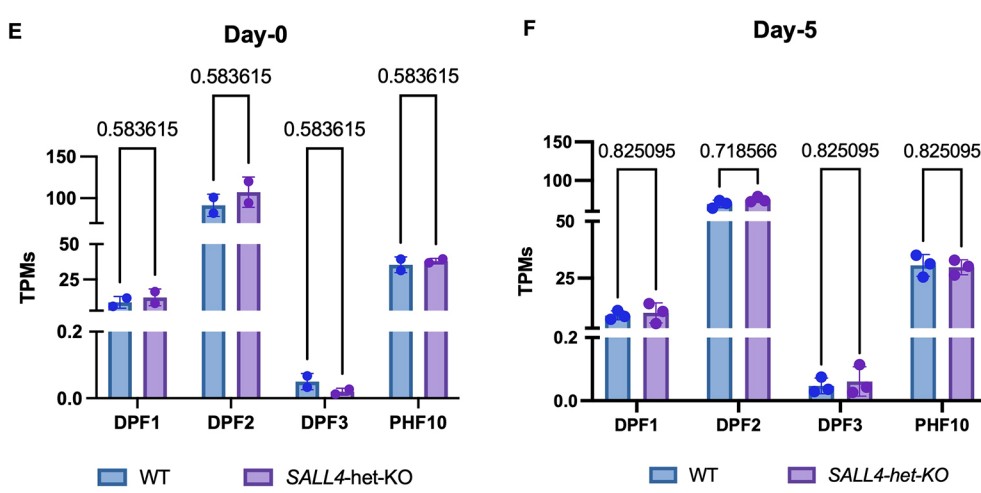

**Fig. 1. SALL4A interacts with the BAF complex through DP+F2.** (A) Immunoblot of nuclear extract (5% input), co-immunoprecipitations (co-IP) of ARID1B and IgG (negative control) from *SALL4*-WT cells at day 5. (B) AlphaFold structure model showing the predicted interactions between SALL4A (red), DPF2 (dark blue) and other BAF components, including SMARCA4, SMARCB1 and ACTL6A. (C) SALL4A protein sequence and visual representation of output model of SALL4A (model 0). C2H2 zinc-fingers (ZFs) annotations are highlighted in dark purple and numbered. Interactions between SALL4 and DPF2 through the ZFs 4, 5 and 6 (ZFC3) and 7 (ZFC4) are highlighted in light purple. Made with BioRender.com. (D) Zoomed view of SALL4 (pink) and DPF2 (purple) interacting surfaces. Highlighted are the functional domains involved in the predicted interaction. In red, contacts predicted by AlphaFold (Jumper et al., 2021). (E) Expression (transcripts per million; TPMs) of the four members of the DPF protein family in iPSCs (day 0; *n*=2). Differences between conditions were assessed using unpaired Student's *t*-test. (F) Expression of the four members of the DPF protein at day 5 (*n*=3). Differences between conditions were assessed using unpaired student's *t*-test. Data are mean±s.e.m. Individual biological replicates are displayed as circles.

BAF is a multi-subunit assembly composed of a dozen subunits, which are differentially incorporated in a tissue-specific manner (Hodges et al., 2016). Based on which subunit is present in each complex, in mammals these can be classified into canonical BAF, polybromo-associated BAF and non-canonical BAF (Zhang et al., 2021). While our MS and co-IP experiments supported the interaction between SALL4 and canonical BAF (ARID1B is not incorporated in other BAF configurations), neither of these two approaches has the resolution to identify the exact BAF subunit that mediates the interaction with SALL4. Therefore, we performed an *in silico* analysis using AlphaFold3 (Jumper et al., 2021). We tested various BAF subunit combinations along with four histones and an AT-rich DNA sequence which has been empirically shown to be recognised by SALL4 (5′-GGAAATATTACC-3′; PDB: 8A4I) and SALL4A. The output was statistically significant, with a 0.45 predicted template modelling (pTM) score and a 0.47 interface predicted template modelling (ipTM) score (Fig. S1A,B). Through this approach, AlphaFold3 predicted a direct interaction between DPF2, a broadly expressed BAF subunit, and the zinc-fingers 5 and 6 (ZFC3) and 7 (ZFC4) of SALL4A (Fig. 1B-D; Fig. S1A,B; Table S1). *DPF2* is a member of the BAF45 gene family, which also includes *PHF10*, *DPF1* and *DPF3*. We were able to rule out its paralogues based on their distinct expression patterns, incorporation into specific BAF complexes, and previous reports that BAF45 subunits are not functionally redundant (Zhang et al., 2019a). In particular, PHF10 is exclusively associated with the pBAF, which does not incorporate ARID1B (Krasteva et al., 2017), while DPF1 and DPF3 are incorporated in BAF only in neurons (Lessard et al., 2007). Moreover, we compared the gene expression of each DPF paralogue at both day 0 and day 5 and observed no significant difference in expression between wild type (*SALL4*-WT) and *SALL4* haploinsufficient iPSC lines (hereafter *SALL4*-het-KO) for any of them (Fig. 1E,F).

This prediction was particularly interesting given that DPF2 deletion has been linked to early neural ectoderm differentiation in ESCs (Zhang et al., 2019a). Notably, of the two main SALL4 isoforms (SALL4A and SALL4B), SALL4B lacks the portion of exon 2 that contains ZFC3. This prompted us to explore whether the SALL4-DPF2 interaction could be isoform-specific. To address this, we reanalysed the interactions in AlphaFold3, incorporating SALL4B to the same BAF subunits as before. To ensure biological relevance, we first excluded two of the five models that did not exhibit interactions with the AT-rich DNA sequence, as SALL4 is expected to bind DNA for its function. Among the remaining three models, we observed that SALL4B interacted with DPF2 but, unlike the SALL4A isoform, its interaction was not domain specific. Instead, the predicted interactions for SALL4B were characterised by annotated disordered regions and appeared to be largely driven by a non-specific, zinc-finger-to-zinc-finger association (Fig. S1C,D; Table S2), with no particular domain-specific interactions identified.

To accurately interpret our AlphaFold3 predictions, we examined two key confidence metrics: the predicted aligned error (PAE), which assesses the reliability of the relative positioning of residues within the structure, and the predicted local distance difference test (pLDDT), which provides a per-residue confidence score for the model (Table S1). Our SALL4-DPF2 contact analysis revealed 394 atomic interactions (centre-to-centre distance ≤8 Å) involving 139 residues (Fig. 1D; Table S1). Encouragingly, the pLDDT values were highest in known functional domains involved in the interaction, particularly ZF5, ZF6 and ZF7 in SALL4A, as well as the PHD1 and PHD2 domains in DPF2, reinforcing the reliability of these predicted structures. Consistently, our model also aligned closely with experimentally resolved structures at 3 Å resolution available for the BAF base module (Fig. 1E). To refine our analysis, we focused on residues where at least half of their atomic interactions had a pLDDT>65 (70±10%) and that were annotated as part of a functional domain, ensuring that only higher-confidence and potentially functional interactions were considered. As a result, ZF7 was excluded from further analysis (Table S1).

In summary, our experiments and analyses confirmed that SALL4 interacts with BAF during early stages of human CNCC specification, and predicted that this interaction might be mediated by the BAF subunit DPF2 through ZFC3.

## Generation of a *SALL4*-haploinsufficient hiPSC lines

Our experiments have so far revealed that SALL4 interacts with BAF at early stages of CNCC specification, and specifically at the neuroectodermal stage. To investigate the scope of this interaction, we generated *SALL4*-het-KO (Fig. 2A). We opted for heterozygous knockout (KO) of *SALL4* as this recapitulates the human *SALL4* syndromes, which are all caused by *SALL4* haploinsufficient variants (Al-Baradie et al., 2002; Kohlhase et al., 2002; Terhal et al., 2006). Two different CRISPR/Cas9 clones were generated and used as biological replicates. Specifically, the CRISPR-KO was generated by introducing an 11 bp (clone C4) deletion and a 19 bp (clone F1) deletion, respectively (Fig. S2A), both within the 3′-terminus of exon 1 of *SALL4*. This is a conserved region across both *SALL4* isoforms that lacks functionally significant domains. Therefore, both *SALL4* isoforms were equally impacted by the CRISPR deletions.

SALL4 immunoblot and immunofluorescence assays confirmed that SALL4 protein expression was significantly reduced in the *SALL4*-het-KO iPSCs compared to controls (*SALL4*-WT; Fig. 2B, C; Fig. S2B,C). We noted that SALL4A protein expression in the *SALL4*-het-KO line was significantly less than the expected 50% of the WT, suggesting potential negative feedback mechanisms triggered by the reduced SALL4 levels, possibly due to the preferentially homodimeric nature of this protein. On the other hand, SALL4B protein was barely detectable in both *SALL4*-WT and *SALL4*-het-KO conditions (Fig. 2B).

In contrast, at the gene expression level we did not observe significant differences in *SALL4* expression between *SALL4*-WT and *SALL4*-het-KO, for any isoform (Fig. 2D). This may reflect autoregulatory mechanisms, where reduced SALL4 protein levels in the knockout trigger compensatory transcriptional upregulation of the gene – this does not however result in increased protein level because of nonsense-mediated decay or other post-transcriptional regulatory mechanisms.

Next, we investigated whether other SALL paralogues (*SALL1*, *SALL2*, *SALL3*) could compensate for SALL4 loss. First, we investigated their mRNA expression (Fig. 2E) and found *SALL1* as the only paralogue significantly upregulated in *SALL4*-het-KO (Fig. 2E). SALL1 upregulation was ever greater at protein level (Fig. 2F), indicating a potential compensation in pluripotent cells by post-transcriptional regulation.

Importantly, *SALL4*-het-KO iPSCs remained viable and exhibited hallmarks iPSC morphology (Fig. S2D). Furthermore, these cells showed no significant difference in growth rate compared to controls (Fig. 2G).

## Decreased SALL4 expression does not affect the stem cell identity of the iPSCs

Given the debated role of SALL4 in maintaining pluripotency, we investigated whether *SALL4*-het-KO impacted iPSC stem cell identity. Protein levels of the pluripotency transcription factors

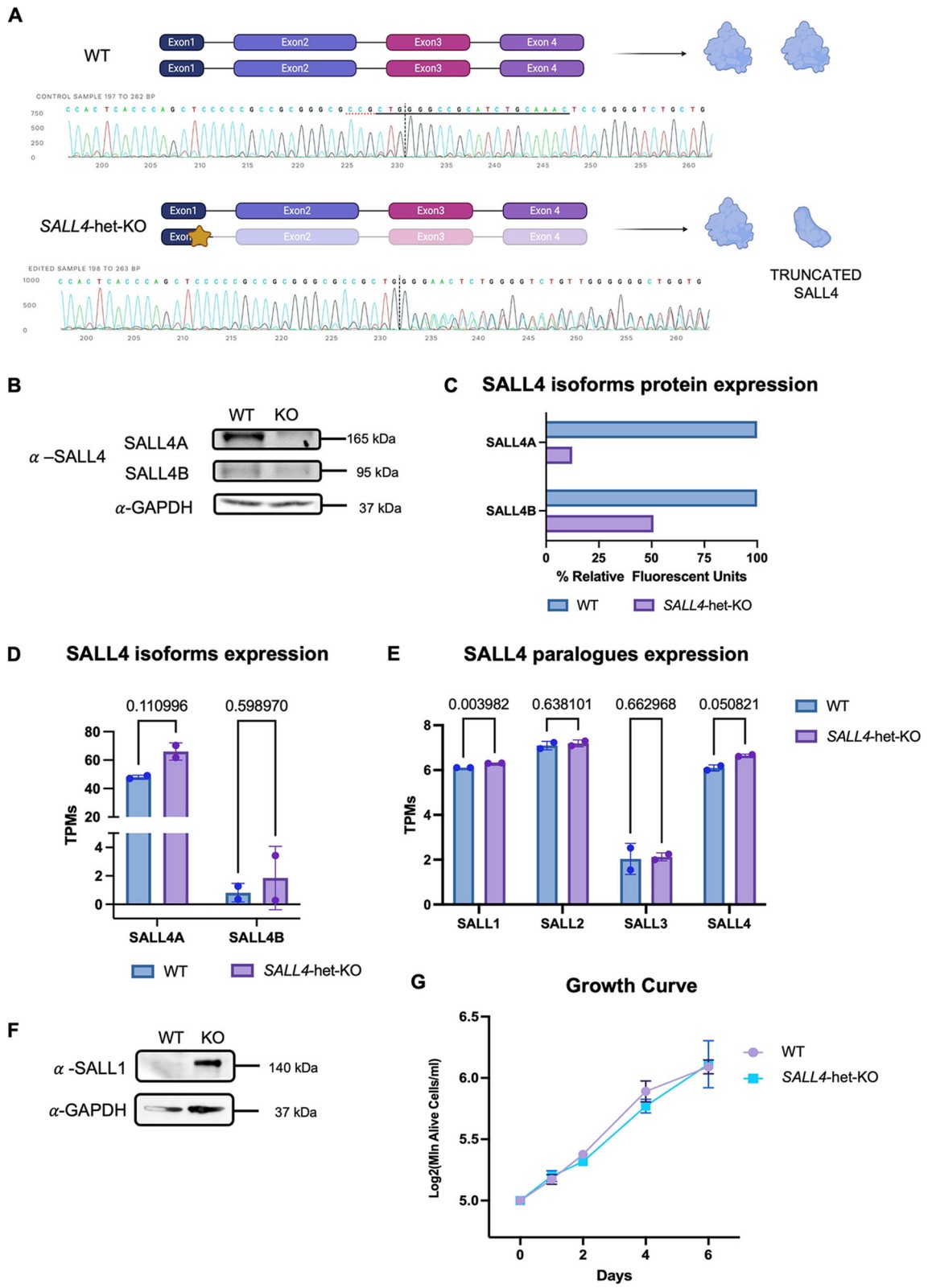

Fig. 2. Generation of a *SALL4*-haploinsufficient hiPSC line. (A) Visual representation of the CRISPR/Cas9 approach used for *SALL4*-het-KO. The star represents the exact location of the deletion. Sanger sequencing confirming the 11-bp deletion in one allele of *SALL4* is displayed (clone F1). Created in BioRender by Demurtas, M., 2025. https://BioRender.com/ufbzgfa. This figure was sublicensed under CC BY 4.0 terms. (B) Immunoblot for SALL4 in *SALL4*-WT and *SALL4*-het-KO (clone F1) iPSC lines. (C) Densitometry analysis of SALL4A and SALL4B expression in WT and *SALL4*-het-KO iPSC lines from immunoblot assay. (D) RNA expression (transcripts per million; TPMs) for SALL4A and SALL4B in WT and *SALL4*-het-KO iPSCs (*n*=2). (E) Bar plots displaying expression TPMs of all *SALL* paralogues in *SALL4*-WT and *SALL4*-het-KO iPSCs (*n*=2). Individual biological replicates are displayed as circles. (F) Immunoblot for SALL1 in *SALL4*-WT and *SALL4*-het-KO (clone F1) iPSC lines. (G) Growth curve comparing *SALL4*-WT with *SALL4*-het-KO iPSC lines. *N*=2 biologically independent experiments. Statistical significance was assessed using unpaired Student's *t*-test. Data are mean±s.e.m.

NANOG and OCT4 remained unaltered in the *SALL4*-het-KO cells relative to controls (Fig. 3A). Similarly, the levels of the pluripotency surface markers TRA-1-60-R and SSEA4 were also not significantly affected by decreased SALL4 (Fig. 3B; Fig. S4A). Moreover, we examined the transcriptome of the iPSCs and identified only 114 differentially expressed genes in *SALL4*-het-KO cells relative to controls (FDR<0.05; Fig. 3C; Table S3). Notably, canonical pluripotency markers, such as *POU5F1*, *NANOG*, *SOX2*, *KLF4*

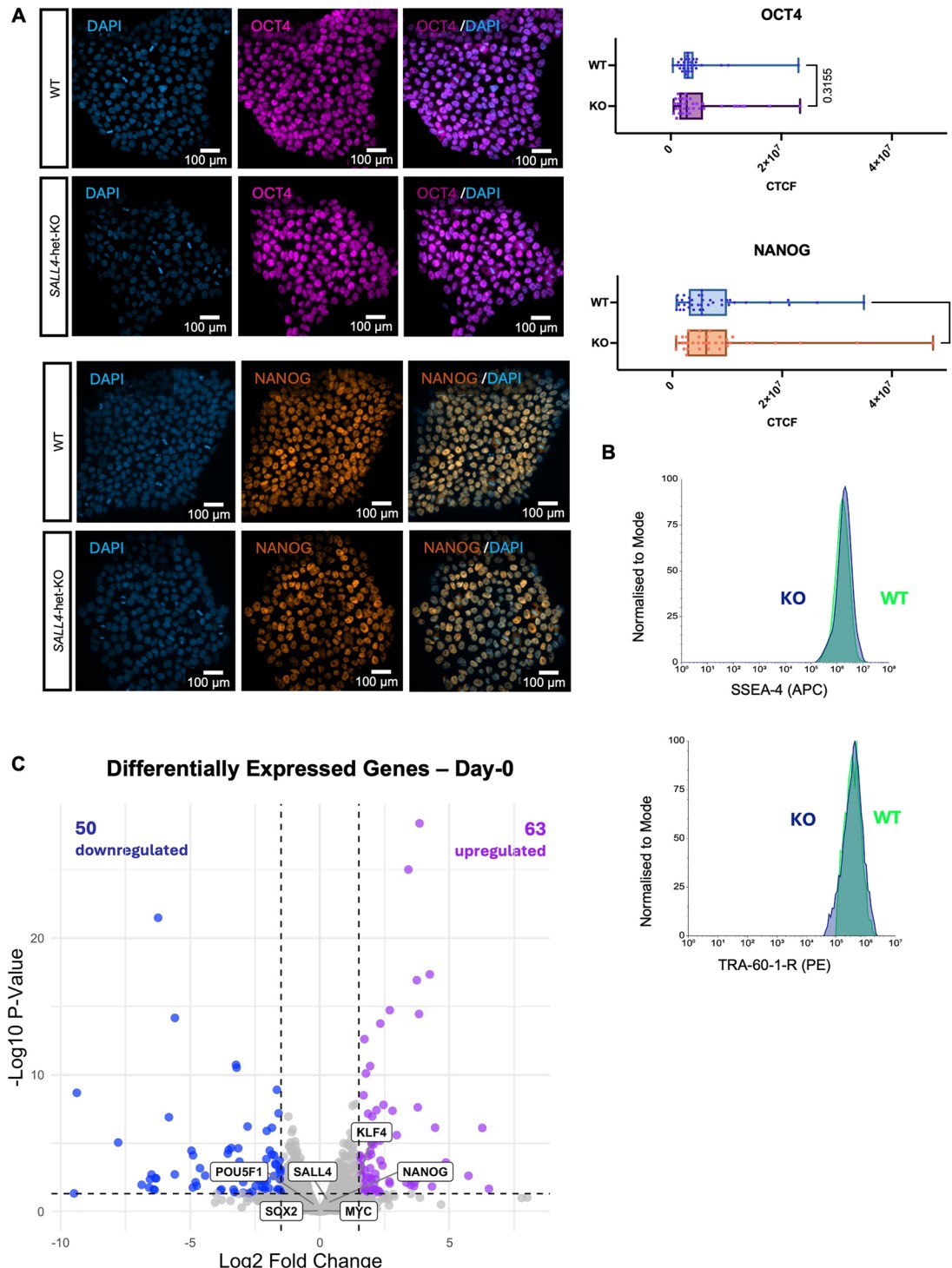

**Fig. 3. Stem cell identity is unaffected by SALL4 loss.** (A) (Left) Immunofluorescence labelling quantifying the expression of the canonical pluripotency markers OCT4 and NANOG in *SALL4*-WT and *SALL4*-het-KO (clone F1) iPSCs. Scale bars: 100 μm. (Right) Corrected total cell fluorescence values were used to quantify and compare marker expression (OCT4, *n*=34; NANOG, *n*=37). Box plots show median values (middle bars) and first to third interquartile ranges (boxes); whiskers indicate minimum and maximum values; dots represent each fragmented nucleus considered. (B) Flow cytometry quantifying the expression of surface markers for pluripotency (TRA-60-1-R and SSEA-4). (C) Volcano plot of genes differentially expressed in *SALL4*-het-KO relative to *SALL4*-WT in iPSCs (day 0). Blue dots (*n*=50) represent downregulated genes with *P*adj<0.05 and log2FoldChange<−1.5. Purple dots (*n*=63) represent upregulated genes with *P*adj<0.05 and log2FoldChange>1.5.

and *MYC*, were not significantly differentially expressed (Fig. 3C; Table S3). Consistent with this, gene ontology analysis confirmed that neither upregulated (Fig. S3A) nor downregulated (Fig. S3B) genes were enriched for pluripotency-associated terms. Finally, by performing tri-lineage differentiation we confirmed that SALL4 depletion did not impact the ability of the *SALL4*-het-KO cells to successfully differentiate into all three of the germ layers (Fig. S3C). Together, these findings suggest that *SALL4* haploinsufficiency did not affect iPSC stemness and self-renewal, possibly as a result of SALL1-mediated compensation.

### SALL4 relocates to CNCC developmental enhancers upon CNCC induction cues

After establishing that *SALL4*-het-KO did not affect iPSC stemness, we differentiated the iPSCs into CNCCs using the same aforementioned protocol (Bajpai et al., 2010; Barnada et al., 2024; Mitchell et al., 2025 preprint; Pagliaroli et al., 2021; Prescott et al., 2015) (Fig. 4A). Since the SALL4-BAF interaction was detected at day 5 (Pagliaroli et al., 2021), corresponding to neuroectodermal spheres, we performed CUT&RUN to profile SALL4 binding in *SALL4*-WT cells in iPSCs and at day 5 of differentiation. We performed two SALL4 CUT&RUN replicates per time point, and only retained replicated peaks in each time point (FDR<0.05). With this approach, we identified comparable amounts of SALL4-bound genomic sites at day 0 and day 5 (29,088 and 32,707, replicated peaks, respectively; FDR<0.05; Fig. 4B). Further analysis of these sites indicated that ~20% of the day 0 SALL4 peaks were unique to iPSCs and not detected at day 5 (hereafter SALL4 day-0-specific peaks; *n*=6114; Fig. 4B,C, left). On the other hand, ~50% of the replicated peaks detected at day 5 were exclusive of this developmental stage and were not detected in iPSCs (hereafter SALL4 day-5-specific peaks, *n*=17,137; Fig. 4B,C). Finally, a subset of ~15,000 sites remained bound by SALL4 at both time points. These data indicate that, upon neural crest differentiation cues, there is a major shift in SALL4 binding, resulting in the relocation of this transcription factor at ~17,000 previously unbound regions.

To characterise the scope of SALL4 relocation, we analysed the positional distribution of SALL4 peaks by associating each peak with its nearest transcription start site (TSS), identifying 4362 genes at day 0 and 6861 genes at day 5 (Table S4). Consistent with the established role of SALL4 as an enhancer-binding transcription factor, 87% of SALL4-bound regions at day 0 were located >1 kb from the nearest TSS, suggesting that these represent putative enhancers. This proportion increased substantially to 97% at day 5 (Fig. 4D). GO terms enriched in the 'day-5-specific' genes included pathways associated with neural crest development, such as neural tube development, face development and cardiac chamber development (Fig. 4E; FDR<0.05).

To further characterise the SALL4 bound elements at day 0 and day 5, we performed sequence-based computational motif analysis. At day 0, motifs associated with core pluripotency transcription factors such as OCT4, NANOG and SOX2 did not rank among the top 20 enriched sequences, further supporting that SALL4 may not be essential for maintaining pluripotency in human cells (Fig. S5A).

On the other hand, in addition to general enhancer activator factors (AP-1, TEAD), the day-5-specific SALL4 peaks were enriched for many CNCC signature factors, including SOX9, TFAP2A, TWIST1/2, SNAI1, NR2F1/2, TCF4/12 and several others (Fig. 4F). Moreover, neuroectodermal transcription factors, such as ZIC2/3 were also enriched. Additionally, *de novo* motif analysis in iPSCs revealed enrichment for the CCCTC-binding factor (CTCF)-like motifs (Fig. S5B), while at day 5 there was enrichment for enhancer activators TEADS/ETS and AP-1 (Fig. S5C). Finally, AT-rich motifs resembling the SALL4 annotated motif were found only in a small minority of peaks in iPSCs (Fig. S5B). This might suggest that SALL4 binding at the CNCC enhancers is not dictated by recognition of its cognate motif, and is possibly mediated by other lineage-specific transcription factors.

In summary, our data so far indicate that SALL4 relocates to CNCC developmental enhancers upon differentiation cues.

### Chromatin accessibility at the CNCC enhancers is SALL4 dependent

Given that we demonstrated that SALL4 interacts with BAF at early stages of CNCC specification, we reasoned that SALL4 binding at the CNCC developmental enhancers could be necessary for their chromatin activation. We profiled chromatin accessibility using ATAC-seq at day 0 and day 5 in both *SALL4*-WT and *SALL4*-het-KO conditions (two biological replicates per time point per condition, only peaks present in both replicates were considered; FDR<0.05). The number of ATAC-seq replicated peaks identified in the *SALL4*-WT condition was comparable between day 0 (61,660 peaks; FDR<0.05) and day 5 (58,375 peaks; FDR<0.05; Fig. 5A). Next, we specifically investigated the chromatin state of the 17,137 genomic sites that gained SALL4 binding at day 5 and found that SALL4 binding at these sites correlates with a marked increase in chromatin accessibility compared to day 0 (Fig. 5B). Strikingly, the accessibility at these regions was significantly attenuated in *SALL4*-het-KO relative to *SALL4*-WT (Fig. 5C). This suggests that, in response to differentiation signals, SALL4 relocates to CNCC developmental enhancers, which subsequently become activated through chromatin opening, which is SALL4 dependent.

Motif analysis of regions that lost accessibility in *SALL4*-het-KO cells (*n*=1649; Fig. 5A) showed again enrichment for CNCC factors such as ZIC2, ZEB1/2 and SNAIL, along with BMP signalling factors such as SMADs and enhancer activators such as AP-1 (Fig. 5D). On the other hand, regions aberrantly accessible in the *SALL4*-het-KO cells (*n*=3460; Fig. 5A) showed enrichment for pluripotency factors, CTCF and neural plate markers (Fig. 5E).

### SALL4 recruits the BAF complex at CNCC developmental enhancers to increase chromatin accessibility

Since SALL4 binding at the CNCC enhancers is necessary for chromatin opening, we investigated whether the SALL4-bound developmental enhancers were also occupied by BAF, and if BAF binding at these sites could be SALL4 dependent. We performed chromatin immunoprecipitation sequencing (ChIP-seq) for BRG1 (SMARCA4), which is the only catalytic BAF subunit active at this stage, in *SALL4*-WT cells at day 5. We found that BRG1 consistently binds the same genomic sites as SALL4, supporting the idea that these factors interact and co-occupy regulatory elements during early differentiation (Fig. 6A).

To assess whether BRG1 binding depends on SALL4, we performed ChIP-seq for BRG1 in both *SALL4*-WT and *SALL4*-het-KO cells at day 5. In *SALL4*-WT, we detected 32,833 replicated BRG1 peaks (FDR<0.05), as opposed to only 14,116 peaks in *SALL4*-het-KO. This suggests that SALL4 loss triggered widespread loss of BAF from chromatin. We focused on BRG1 peaks lost in *SALL4*-het-KO cells and found that these sites were largely not bound by SALL4 at day 0, but they gained SALL4 binding at day 5 (Fig. 6B-D). As expected, loss of SALL4 and BRG1 from these sites led to marked decrease of chromatin accessibility (Fig. 6E). Moreover, we explored the identity of some of these putative enhancers by confirming that these overlapped with some of those reported on databases such as

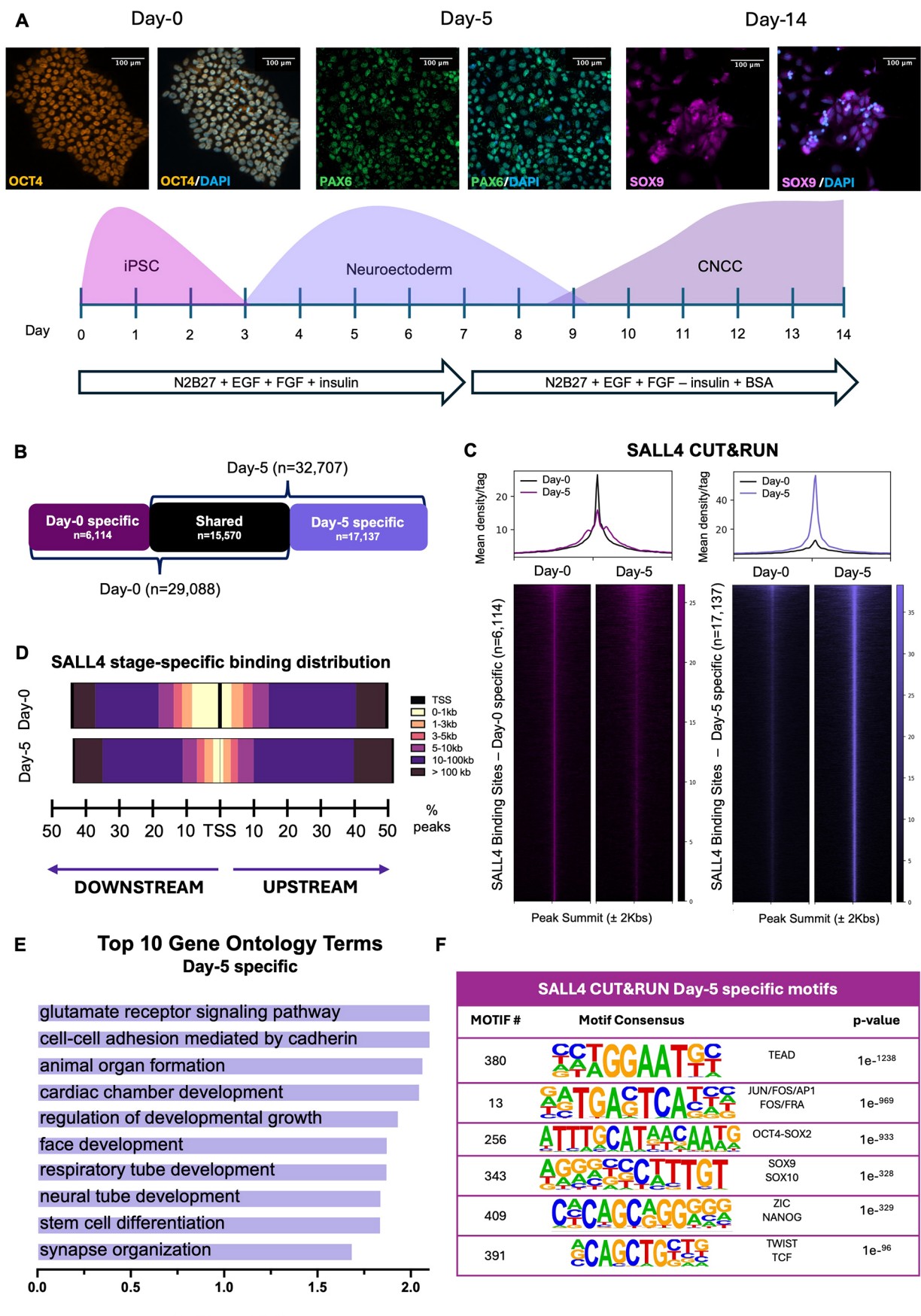

Fig. 4. See next page for legend.

**Fig. 4. SALL4 relocates to CNCC enhancers to induce chromatin accessibility.** (A) (Bottom) Schematic representation of the iPSC-to-CNCC differentiation protocol. The experimental timeline shows *in vitro* differentiation of iPSCs into CNCCs via a neuroectodermal-like stage. (Top) Immunofluorescence labelling quantifying the expression of the canonical stage-specific markers (iPSCs: OCT4; neuroectoderm: PAX6; CNCCs: SOX9) showing ratio of cells successfully undergoing stage-specific specifications during CNCC generation. Scale bars: 100 μm. (B) Diagram showing the number of regions bound by SALL4 uniquely at day 0 ('day-0-specific', *n*=6114), at day 5 ('day-5-specific', *n*=17,137) and regions that maintain SALL4 binding at both time points (shared, *n*=15,570). (C) Heatmaps showing the distribution of SALL4 binding at day 0 and day 5 across day-0-specific (left) and day-5-specific (right) binding regions. CUT&RUN signal intensity was plotted across ±2 kb regions centred on peak summits. Rows represent individual peaks ranked by signal intensity. (D) Genomic distribution of SALL4 binding sites relative to the nearest transcription start site (TSS). Peaks are categorised by distance into six bins: <1 kb, 1-3 kb, 3-5 kb, 5-10 kb, 10-100 kb and >100 kb. Peaks located at >1 kb from the nearest TSS were classified as putative enhancers. Percentages of peaks for each bin are represented by bars to the left (downstream) or right (upstream) of the TSS. (E) Gene ontology analysis of downregulated nearest genes to day-5-specific SALL4-bound regions as determined using WebGestalt (Liao et al., 2019) over-representation pathway analysis, using affinity propagation as parameter for redundancy removal (FDR<0.05). (F) Table of motifs enriched in day-5-specific SALL4-bound regions as determined using HOMER (Heinz et al., 2010).

VISTA Enhancer Browser (Kosicki et al., 2025) and GeneHancer (Fishilevich et al., 2017; Fig. 6F). Overall, these experiments support that SALL4 is required to recruit BAF at CNCC enhancers.

To further confirm that these regions are enhancers, we leveraged publicly available H3K4me1 and H3K4me3 ChIP-seq datasets generated using a comparable protocol at our day 5 equivalent (Prescott et al., 2015; Rada-Iglesias et al., 2011). Moreover, we performed ChIP-seq for H3K27ac at day 5, day 11 and day 14, roughly corresponding to CNCC-induction, early CNCC and late CNCC stages, respectively. This analysis revealed that nearly all the sites bound by SALL4 and BRG1 were marked by the enhancer mark H3K4me1, and only a small subset of them had the promoter mark H3K4me3 (Fig. S6A). Moreover, nearly all these sites had the active enhancer mark H3K27ac at day 5, and the acetylation levels progressively increased from CNCC induction to CNCC stage, supporting that these are bona fide CNCC enhancers that increase in activation after SALL4 binding (Fig. S6B,C).

While a large portion of BRG1 peaks was lost in the absence of SALL4, we also identified a subset of novel BRG1 peaks that were absent in *SALL4*-WT cells but gained in the *SALL4*-het-KO cells. These BRG1 peaks gained in the *SALL4*-het-KO cells were associated with genes linked to neurogenesis and nervous system development (Fig. S7A; Table S4). In contrast, the genes located near BRG1 peaks lost in the *SALL4*-het-KO cells were involved in embryonic organ development, epithelial differentiation and cell migration processes (Fig. S7B; Table S4).

Together, these results suggest that SALL4 is a regulator of BAF recruitment to CNCC enhancers, ensuring their accessibility during differentiation. In the absence of SALL4, BRG1 fails to localise to these sites and is partially redirected to neurodevelopmental enhancers, leading to a misallocation of chromatin remodelling, and to a shift towards default neural differentiation at the expense of CNCC induction and specification.

### SALL4-het-KO cells are not able to exit the neural ectoderm stage and to induce the cranial neural crest fate

Our data thus far indicate that SALL4 relocates to CNCC enhancers as early as the neuroectodermal stage, recruiting BAF to these

regulatory elements for chromatin activation. Next, we investigated the functional consequences of SALL4 loss by generating CNCCs using *SALL4*-WT and *SALL4*-het-KO cells. As expected, *SALL4*-WT CNCCs expressed the signature CNCC markers at both the mRNA (Fig. 7A) and protein (Fig. 7B; Fig. S4B) level, while *SALL4*-het-KO failed to induce these genes (Fig. 7A). This was further supported by RNA-seq conducted at day 14, which identified 1106 genes differentially expressed in *SALL4*-het-KO CNCCs relative to *SALL4*-WT (FDR<0.05; Fig. 7C; Table S3). This was in sharp contrast with the small number of genes differentially expressed at the iPSC stage (*n*=114; Fig. 3C; Table S3) and at day 5 (*n*=170; Fig. S7C; Table S3).

Of the 1106 genes differentially expressed at day 14, ~60% (*n*=659) were downregulated while the remaining were upregulated (Fig. 7C). Canonical CNCC markers such as *TFAP2A*, *NR2F1* and *SOX9* were significantly downregulated in *SALL4*-het-KO cells. Additionally, genes associated with neural plate border, early neural crest induction and epithelial-to-mesenchymal transition (EMT) were also downregulated (Fig. 7D,F). In line with this, gene ontology analysis revealed that pathways associated with face development, autonomic nervous system, gastrulation and regulation of morphogenesis were all downregulated (Fig. 7D). The 447 upregulated genes were instead involved in cell regulatory pathways (Fig. 7E).

As aforementioned, genes associated with EMT were dysregulated. The process of EMT requires a switch in cadherin expression where class-1 cadherins must be downregulated while class-2 cadherins must be upregulated. Related to this, our RNA-seq data revealed that this cadherin switch did not occur in the differentiating *SALL4*-het-KO iPSCs, suggesting that the cells remained in an epithelial state. This was further corroborated by the upregulation of several epithelial genes in the differentiating *SALL4*-het-KO cells, including *EPCAM*, *CDH1* and *OVOL2* (Fig. 7F). The latter is involved in maintaining epithelial lineages by repressing the expression of the EMT gene *ZEB1*, which was downregulated in the *SALL4*-het-KO condition (Haensel et al., 2019; Liu et al., 2018a).

In summary, our data indicate that CNCC specification failed in *SALL4*-het-KO conditions, as reflected by the large number of differentially expressed genes. On the other hand, the transcriptomes of iPSCs and neuroectodermal cells (i.e. day 5) were only modestly affected by SALL4 loss, suggesting that SALL4 may not be essential for neuroectoderm formation, while it is essential for CNCC induction and specification.

To investigate this further, we systematically examined the expression of a panel of stage-specific marker genes at day 0, day 5 and day 14 (Fig. S8A-D) including core pluripotency genes (*POU5F1* and *NANOG*), neural plate border specifiers (*ZIC1, GBX2, MSX1/2, PAX3/7, GATA2/3*), CNCC specification markers (*SNAI1/2, SOX9, TFAP2A*), and post-migratory CNCC markers (*SOX5, SOX6, SOX10*). In *SALL4*-WT cells, pluripotency markers exhibited a progressive and expected downregulation as differentiation advanced from day 0 to day 14. In *SALL4*-het-KO cells, a similar decline in pluripotency markers was observed between day 0 and day 5, but instead of continuing to decrease, these markers remained similarly expressed through day 14 (Fig. S8A). Despite this, the two conditions showed similar trends, confirming that decreased levels of SALL4 did not significantly impact pluripotency marker expression during exit from pluripotency. Similarly, no significant difference was observed in the expression of neural plate border specifiers (Fig. S8B), suggesting that SALL4 is not required for human neuroectoderm specification. Conversely, CNCC specification markers and post-migratory CNCC markers all increased in expression after day 5 in *SALL4*-WT cells, while the SALL4-depleted cells failed to maintain a

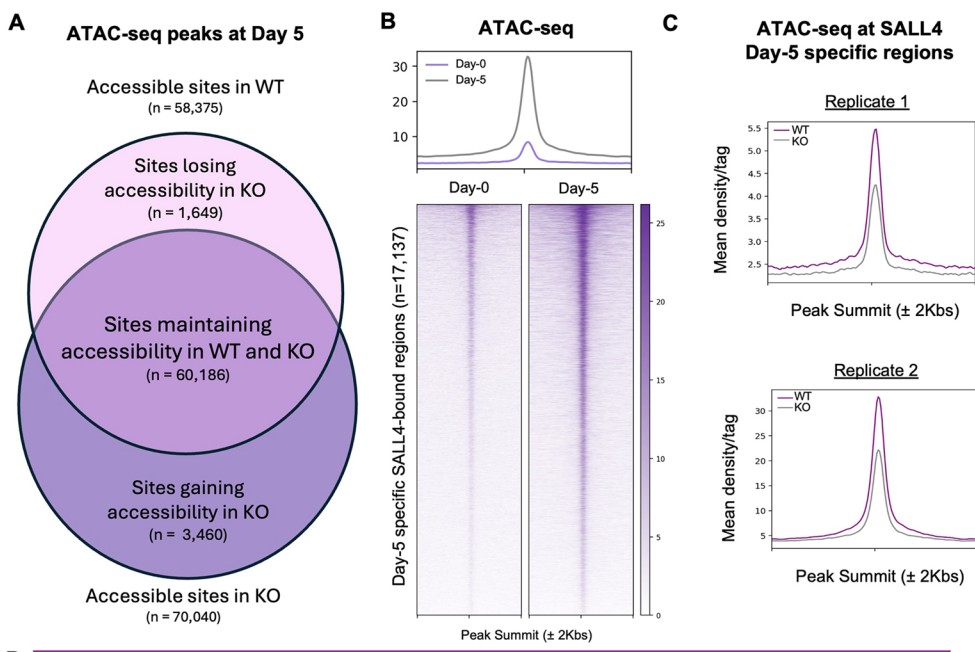

**Fig. 5. Chromatin accessibility at the CNCC enhancers is SALL4 dependent.** (A) Venn diagram showing chromatin accessibility in *SALL4*-WT (ATAC-seq peaks, n=58,375) and *SALL4*-het-KO (n=60,186). It also depicts the number of regions uniquely accessible in *SALL4*-WT ('Sites losing accessibility in KO', n=1649), in *SALL4*-het-KO ('Sites gaining accessibility in KO', n=3460) and regions that remain accessible in both conditions ('Sites maintaining accessibility in WT and KO', n=60,186). (B) Heatmaps showing ATAC-seq signals at day 0 (left) and day 5 (right), at day-5-specific SALL4 CUT&RUN peaks. Signal intensity was plotted across ±2 kb regions centred on peak summits. Rows represent individual peaks ranked by signal intensity. (C) Average profiles showing ATAC-seq signal at day-5-specific SALL4 CUT&RUN peaks (n=17,137) in *SALL4*-WT and *SALL4*-het-KO cells in two distinct biological replicates. (D,E) Tables of motifs enriched in regions gaining or maintaining accessibility (D) or regions losing accessibility (E) at day 5 in *SALL4*-het-KO regions as determined using HOMER (Heinz et al., 2010).

comparable increase in expression (Fig. S8C,D). In summary, *SALL4*-het-KO cells are competent to generate neuroectoderm-like cells but fail to induce the neural crest lineage.

### SALL4A but not SALL4B is necessary for CNCC specification

Given previous reports suggesting distinct roles for the two SALL4 isoforms in ESCs (Rao et al., 2010), and our prediction that the interaction between SALL4 and DPF2 may be isoform-specific, we investigated whether CNCC differentiation exhibits preferential dependence on a particular SALL4 isoform.

Time-course mRNA expression in *SALL4*-WT cells revealed that *SALL4B* is consistently lowly expressed throughout differentiation, whereas *SALL4A* expression increases upon differentiation cues, peaking around day 5 and then declining over time (Fig. S9A,B). On the other hand, the paralogue SALL1 is expressed at low levels in iPSCs and barely detectable throughout the entire differentiation protocol (Fig. S9C).

To investigate potential isoform-specific requirements for CNCC development, we selectively degraded SALL4A while preserving SALL4B expression. To achieve this, we treated cells with

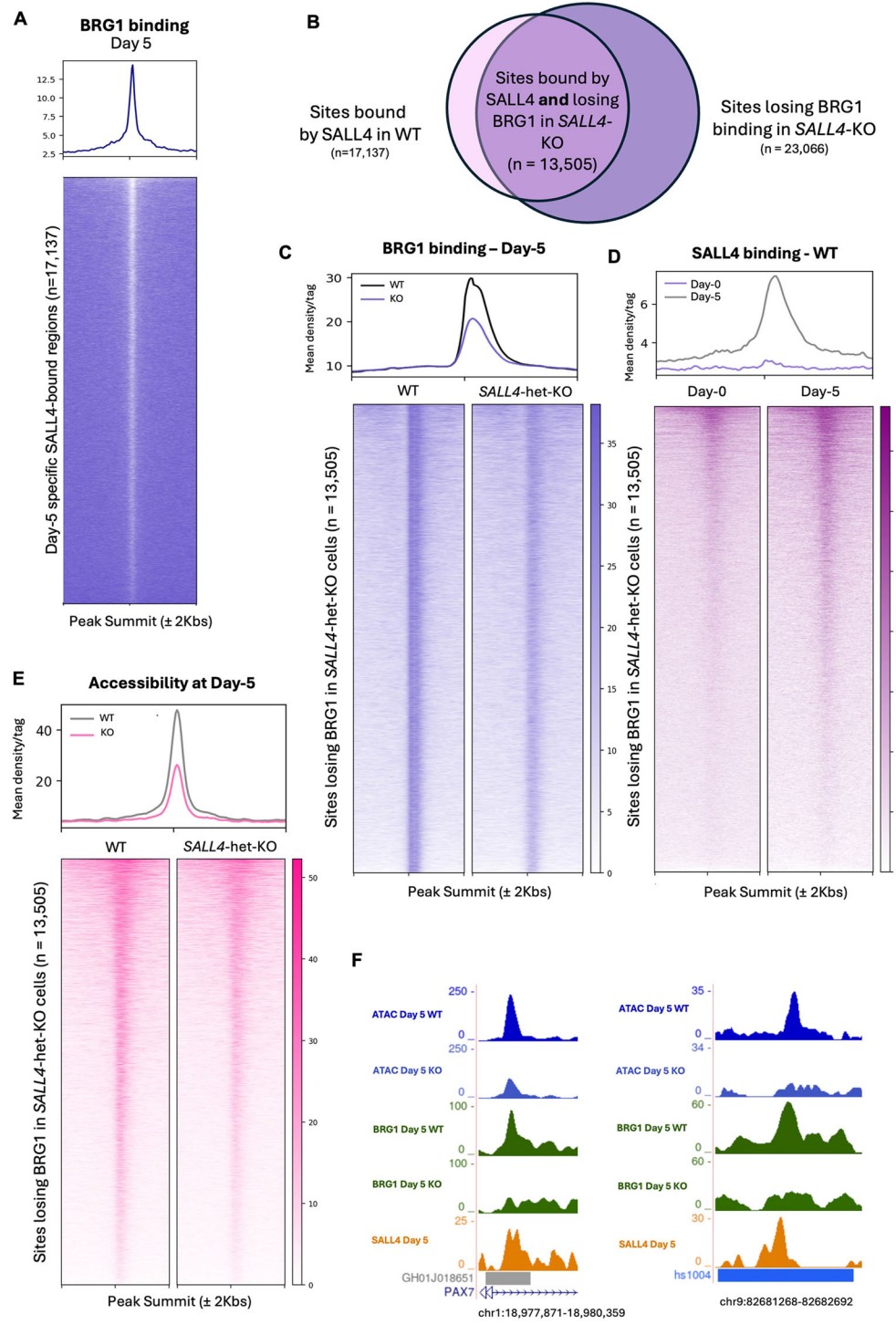

**Fig. 6. SALL4 recruits BAF to CNCC enhancers.** (A) Heatmap showing BRG1 ChIP-seq signal at the genomic sites bound by SALL4 in WT cells at day 5 (*n*=17,137). Signal intensity was plotted across ±2 kb regions centred on peak summits. Rows represent individual peaks ranked by signal intensity. (B) Venn diagram showing peaks bound by SALL4 uniquely at day 5 and that lose accessibility in *SALL4*-het-KO. (C) Heatmap and average profiles show BRG1 ChIP-seq signal at day 5 at the BRG1 peaks lost in *SALL4*-het-KO (clone F1; *n*=13,505). Signal intensity was plotted across ±2 kb regions centred on peak summits. Rows represent individual peaks ranked by signal intensity. (D) Heatmap and average profiles show SALL4 CUT&RUN signal at the genomic sites that lose BRG1 in *SALL4*-het-KO cells at day 0 and day 5 (*n*=13,505). (E) Heatmaps and average profile show ATAC-seq signal at the genomic sites that lose BRG1 in *SALL4*-het-KO cells (clone F1; *n*=13,505). Signal intensity was plotted across ±2 kb regions centred on peak summits. Rows represent individual peaks ranked by signal intensity. (F) Two representative examples of enhancers bound by SALL4 at day 5 that lose BRG1 co-occupancy and chromatin accessibility in the *SALL4*-het-KO (clone F1) cells. Both correspond to ChIP-seq peaks annotated as enhancer regions, as reported in GeneHancer (left) and VISTA Enhancer Browser (right).

pomalidomide, a thalidomide derivative that selectively degrades SALL4A but not SALL4B (Anh Vu et al., 2023 preprint; Liu et al., 2024 preprint; Fig. 8A).

To test the effects of SALL4A degradation, we differentiated *SALL4*-WT iPSCs into CNCCs. Cells were treated with either DMSO (control) or 5 µM pomalidomide (5 µM POM) every 24 h (Fig. 8B). To confirm consistent degradation of SALL4A throughout the differentiation process, we performed immunoblotting at key time points (day 5 and day 14). As expected, SALL4A was successfully degraded, while SALL4B expression remained intact (Fig. 8C).

Importantly, pomalidomide degrades several proteins beyond SALL4, and its protein degradation activity is typically cell-type specific (Sievers et al., 2018). Two of the known pomalidomide targets have suggested functions in neural crest development. These are ARID2 (Bajpai et al., 2010; Yamamoto et al., 2020) and ZMYM2 (Jourdeuil et al., 2025; Renneville et al., 2021). Crucially, immunoblot analysis for these two proteins performed at day 5 following treatment with 5 µM pomalidomide revealed that the levels of both ARID2 and ZMYM2 remained unchanged compared to the DMSO control (Fig. S9D,E), further supporting that pomalidomide

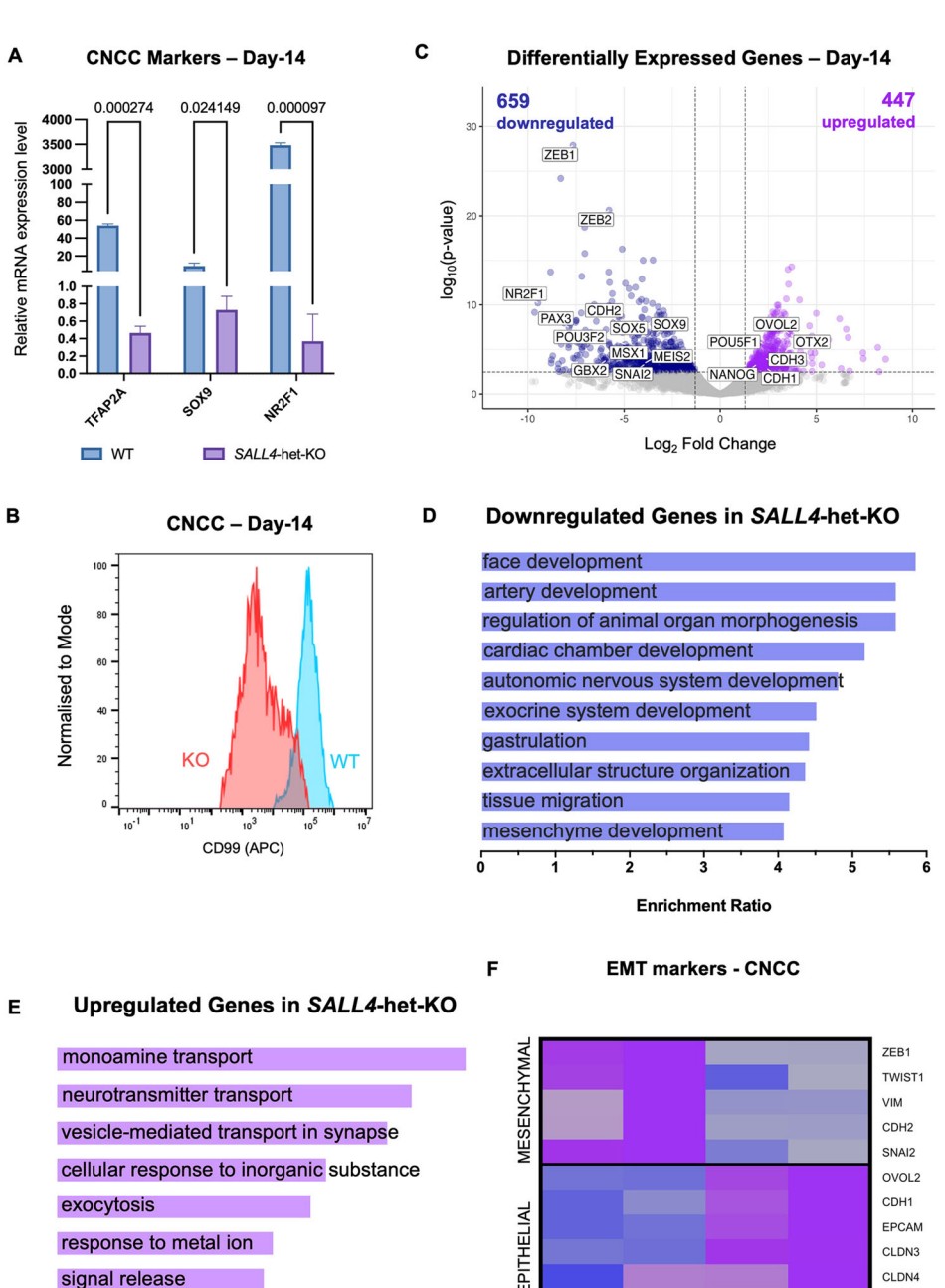

**Fig. 7. CNCC induction and specification are impaired in *SALL4*-het-KO.** (A) Relative mRNA expression of canonical CNCC markers in *SALL4*-WT and *SALL4*-het-KO cells at day 14 (*n*=2). Statistical significance was assessed using unpaired Student's *t*-test. 18S rRNA was used to normalise. Data are mean±s.e.m. (B) Flow cytometry for the CNCC surface marker CD99 in *SALL4*-WT and *SALL4*-het-KO (clone F1) CNCCs at day 14. (C) Volcano plot of genes differentially expressed in *SALL4*-het-KO relative to *SALL4*-WT in CNCCs (day 14). Blue dots (*n*=659) represent downregulated genes with *P*adj<0.05 and log2FoldChange<−1.5. Purple dots (*n*=447) represent upregulated genes with *P*adj<0.05 and log2FoldChange>1.5. (D, E) GO term enrichment for downregulated (D) and upregulated (E) genes in *SALL4*-het-KO cells (FDR<0.05). (F) Heatmap displaying the expression of canonical epithelial-to-mesenchymal transition (EMT) markers in *SALL4*-WT and *SALL4*-het-KO cells (day 14).

affinity for its targets is cell-type specific, as previously suggested (Bajpai et al., 2010; Yamamoto et al., 2020). This control experiment allowed us to rule out potential off-target effects caused by unwanted degradation of one of these two proteins. The other known pomalidomide targets, including IKAROS, AIOLOS, PLZF and ZNFX1 have no established contribution to neural crest specification and in fact they are not highly expressed in our system.

Next, we assessed the downstream effects of SALL4A degradation by performing real-time quantitative polymerase chain reaction (RT-qPCR) and immunofluorescence for key neural crest markers, such as

TFAP2A, NR2F1 and SOX9 (Fig. 8D,E), and EMT markers VIM and EPCAM (Fig. 8F). Degradation of SALL4A recapitulated what we previously observed in *SALL4*-het-KO cells. Specifically, the cells treated with pomalidomide failed to differentiate successfully into CNCCs despite the persistent expression of SALL4B.

Taken together, our results highlight a specific and essential role for SALL4A in orchestrating chromatin dynamics during CNCC specification, and suggest that SALL4B might function in a different regulatory capacity rather than acting as a compensatory factor for SALL4A.

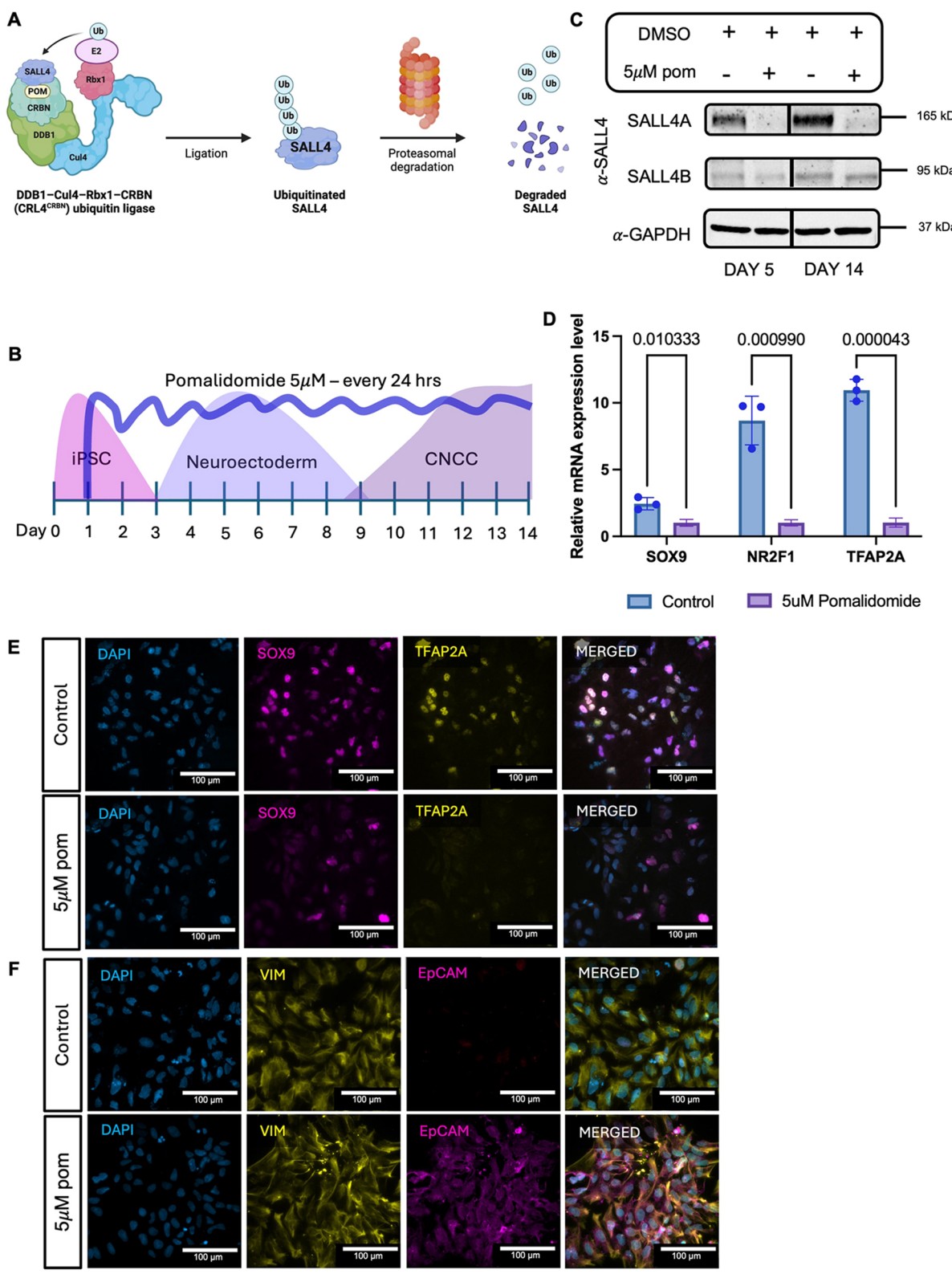

**Fig. 8. SALL4A is the only SALL4 isoform necessary for CNCC specification.** (A) Schematic of pomalidomide-mediated ubiquitin ligation. Ub, ubiquitin; POM, pomalidomide. CRBN, DDB1, Cul4 and Rbx1 are subunits of CRL4/CRBN E3 ubiquitin ligase. Created in BioRender by Demurtas, M., 2025. https://BioRender.com/oij5pkv. This figure was sublicensed under CC BY 4.0 terms. (B) Schematic representation of pomalidomide treatment during iPSC-to-CNCC differentiation protocol. (C) Time-course immunoblot for SALL4 in *SALL4*-WT at day 5 and day 14 after daily treatment with DMSO (control) or 5 µM pomalidomide (5 µM POM). (D) Relative mRNA expression of canonical CNCC markers (SOX9, TFAP2A and NR2F1) in *SALL4*-WT at day 14 treated with 5 µM pomalidomide or DMSO (control) daily for 14 days. Individual biological replicates (*n*=2) are displayed as circles. Statistical significance was assessed using unpaired Student's *t*-test. Data are mean±s.e.m. (E) Immunofluorescence labelling quantifying the expression of the canonical CNCC markers SOX9 and TFAP2A in *SALL4*-WT treated with 5 µM pomalidomide or DMSO (control) daily for 14 days. (F) Immunofluorescence labelling quantifying the expression of the canonical EMT markers EPCAM and VIM in *SALL4*-WT treated with 5 µM pomalidomide or DMSO (control) daily for 14 days. Scale bars: 100 µm.

## DISCUSSION

Mutations in *SALL4* are associated with congenital syndromes which frequently include distinct craniofacial anomalies (Kohlhase et al., 2005; Miller, 1991), suggesting disrupted CNCC development. Notably, at the murine embryonic stage E9.5, Sall4 expression persists within the frontonasal prominence (Tahara et al., 2019), implying a potential role for SALL4 in craniofacial development. Given the limited existing knowledge about SALL4 function in this context, we used a heterozygous *SALL4* knockout hiPSC line to investigate its role CNCC specification.

At early stages of CNCC induction, SALL4 relocates to enhancers of genes involved in CNCC formation, and SALL4 binding at these regulatory elements is required for their activation. In mouse ESCs, ~10% of the total Sall4 protein, and primarily the Sall4A isoform, is incorporated into the NuRD complex (Bode et al., 2016; Miller et al., 2016). Here, we identify and characterise a previously unreported interaction between SALL4 and the BAF complex, predicted to be mediated specifically through the SALL4A isoform and the BAF subunit DPF2. Interestingly, DPF2 and SALL4 exhibit notable functional parallels. Similar to SALL4, DPF2 occupancy undergoes significant repositioning during the transition from pluripotency toward early neural commitment in mouse embryoid bodies (Zhang et al., 2019b). Although the role of DPF2 in CNCCs has not been directly investigated, its early recruitment to neuronal enhancers suggests a potential involvement in neuroectodermal differentiation and lineage commitment of neuroectoderm-derived cells, including CNCCs. Moreover, genomic sequences bound by DPF2 in mouse neuroectoderm are AT-rich, resembling the preferred binding motif of SALL4 (Kong et al., 2021; Nazim et al., 2024; Pantier et al., 2021). It is important to note that AlphaFold3 is built on deep learning rather than reinforcement learning. Consequently, the absence of more comprehensive experimental SALL4 data could significantly affect the reliability of the proposed SALL4–DPF2 interaction model. Additionally, AlphaFold 5000-token limit per prediction poses a constraint when modelling large multi-subunit complexes such as BAF. Fortunately, BAF has been extensively characterised, and our predicted model match experimentally resolved structures at 3 Å resolution.

Our study demonstrated that while CNCC induction fails without SALL4, early neuroectodermal commitment and neural plate border identity remain unaffected. Our findings demonstrate that, during the earliest stages of CNCC specification, SALL4 occupies CNCC-associated enhancers and recruits BAF. Together, these observations support previous studies that proposed that neural crest cell fate commitment is initiated well before overt gene expression changes occur (Basch et al., 2006; Patthey et al., 2008; Stuhlmiller and García-Castro, 2012). This finding is further supported by clinical observations. In fact, while phenotypes arising from *SALL4* haploinsufficiency affect many neural crest derivatives, phenotypes implicating defects in neurodevelopment are exceptionally rare (Borozdin et al., 2004; Ma et al., 2022) and usually associated with dysregulation of potassium channels (Kodytková et al., 2023). In the rare cases of intellectual disability reported, such phenotypes were attributed to broader deletions on chromosome 20q13.13-q13.2 rather than *SALL4* loss alone (Borozdin et al., 2004).

In the absence of SALL4, after 2 weeks of differentiation the cells acquire an aberrant fate, expressing a mix of pluripotency genes (suggesting lack of cell differentiation), and neuronal genes (suggesting derailed cell fate). This indicates that most cells safely reach the neuroectodermal stage but fail to go beyond that, while some of the cells simply do not differentiate or they regress back to pluripotency.

Interestingly, only a small minority of the CNCC enhancers bound by SALL4 has the AT-rich motif that is thought to be recognised by SALL4 (Kong et al., 2021; Pantier et al., 2021). This suggests that SALL4 might bind these enhancers in cooperation with other lineage-specific factors. SALL1 also recognises AT-rich motifs but, similar to SALL4, it also binds unspecifically. Since SALL1 is upregulated in the *SALL4*-het-KO cells, it might possible that some of the sites associated with a gain of BRG1 upon loss of SALL4 could be associated with SALL1 being upregulated and hijacking BAF recruitment to distinct sites.

### Limitations

SALL4 was recently identified as a direct target of thalidomide (Donovan et al., 2018), a drug widely prescribed to pregnant women in the late 1950s to alleviate morning sickness. The subsequent alarming rise in births of children with congenital anomalies, including limb malformations, facial abnormalities such as inner ear defects, ocular motility issues and facial nerve palsy, as well as kidney and heart defects, led to the identification of thalidomide as the causative agent. However, the molecular mechanism by which thalidomide induced these developmental defects remained elusive until nearly six decades later. Building on earlier findings that thalidomide and its derivatives bind to cereblon (CRBN), Matyskiela et al. demonstrated that SALL4 is recruited to the CRL4–CRBN E3 ubiquitin ligase complex following thalidomide-induced ubiquitination, resulting in its proteasomal degradation (Matyskiela et al., 2018). More recently, research demonstrated that only the SALL4A isoform, but not SALL4B, is susceptible to degradation by immunomodulatory imide drugs such as pomalidomide (Anh Vu et al., 2023 preprint). Our data suggest that the SALL4 role in CNCC specification might be isoform-specific and these findings might contribute to shed light on the birth malformations caused by thalidomide. However, while we ensured that known pomalidomide targets important for CNCC developments were not degraded in our system, we cannot rule out degradation of previously uncharacterised and unknown pomalidomide targets with key roles in CNCC specification. While highly unlikely, this is an important limitation of our study, and therefore the findings of our pomalidomide experiments should be taken with caution.

## MATERIALS AND METHODS

### AlphaFold3 structure prediction

FASTA files for each BAF subunit and SALL4 isoforms were collected from the 'Sequence & Isoforms' section of Uniprot (www.uniprot.org). The AT-rich DNA sequences was collected from an empirically reported crystal structure of SALL4 zinc-finger cluster 4 bound to DNA on RCSB PDB (www.rcsb.org, 8A4I; https://doi.org/10.2210/pdb8A4I/pdb). FASTA sequences are listed in Table S2. AlphaFold outputs were downloaded and .cif files were analysed using UCSF ChimeraX-1.9 (https://www.rbvi.ucsf.edu/chimerax). Contacts were found using default parameters and centre-centre distance ≤8 Å.

### Human iPSC culture

WT iPSC lines were obtained from the iPSC Core of the University of Pennsylvania (Control line-1: SV20 line, male, age 43). iPSC lines were cultured in feeder-free, serum-free mTeSR Plus medium (Stemcell Technologies) with 1% Penicillin-Streptomycin (5000 U/ml; Gibco #15140122) on tissue culture plates coated overnight with Geltrex™ LDEV-Free hESC-qualified Reduced Growth Factor Basement Membrane Matrix (Thermo Fisher Scientific). Cells were passaged ~1:8 at 60-80% confluency using 0.5 mM EDTA to create 50-200 cell clusters. For all resources and reagents see Table S5. The study was conducted in accordance with the criteria set by the Declaration of Helsinki.

## CRISPR/Cas9-mediated SALL4 knockout

The heterozygous *SALL4* knockout in the human SV20 iPSC line was generated via CRISPR/Cas9 gene-editing by the biotech company Synthego. Specifically, two clones were used. In both clones Cas9 targeted the first exon of the *SALL4* gene in the established SV20 iPSC line producing an 11-base deletion (c.230delGGGCCGCATCT; clone C4, named KO1 in the manuscript) and a 19-base deletion (c.del226GCTGGGGCCGCATCT; clone F1, named KO2 in the manuscript), which in both cases generated a premature stop codon. The two clones were used as biological replicates in all experiments.

## Trilineage differentiation

iPSCs were differentiated into the three germ layers (endoderm, ectoderm and mesoderm) using the Human Pluripotent Stem Cell Functional Identification Kit (R&D Systems, SC027B). The expression of relevant markers was then assessed using immunofluorescence with 10 µg/ml of antibodies provided in the kit (goat anti-human SOX17, goat anti-human OTX2 and goat anti-human TBXT).

## iPSC-to-CNCC differentiation

All iPSC lines were differentiated into CNCCs as previously described (Prescott et al., 2015). When iPSCs reached ~60% confluence in Geltrex-coated six-well plates, cells were dissociated with accutase in small clusters and were treated with CNCC differentiation media [1:1 Neurobasal medium/D-MEM F-12 medium (Invitrogen), 0.5× B-27 supplement with Vitamin A (50× stock, Gibco, #17504044), 0.5× N-2 supplement (100× stock, Gibco, #17502001), 20 ng/ml human FGF (Gibco), 20 ng/ml recombinant EGF (Sigma-Aldrich), 5 µg/ml bovine insulin (Sigma-Aldrich) and 1× Glutamax-I supplement (100× stock, Invitrogen)]. Medium (2 ml) was changed every other day. At day 7 of differentiation, early migrating CNCCs were transitioned to CNCC early maintenance media until day 14 [1:1 Neurobasal medium/D-MEM F-12 medium (Invitrogen), 0.5× B-27 supplement with Vitamin A (50× stock, Invitrogen), 0.5× N-2 supplement (100× stock, Invitrogen), 20 ng/ml bFGF (BioLegend), 20 ng/ml EGF (Sigma-Aldrich), 1 mg/ml bovine serum albumin, serum replacement grade (Gemini Bio-Products, #700-104P) and 1× Glutamax-I supplement (100× stock, Invitrogen)]. Cell medium (2 ml) was changed every other day. Differentiating cells were passaged with Accutase Cell Detachment Solution (Stemcell Technologies, #07920) for all the subsequent passages when cells reached 80% confluence, plated in fresh, Geltrex-coated six-well plates. After passaging, cells were treated with 0.05% ROCK-inhibitor (Stemcell Technologies, Y27632) for 24 h.

## RT-qPCR

Cells were harvested as single cells, and the total RNA was extracted using Monarch® Total RNA Miniprep Kit (New England Biolabs) following the manufacturer's instructions. We used 600 ng of total RNA to synthesise cDNA using Maxima First Strand cDNA Synthesis Kit for RT-qPCR (K1641, Thermo Fisher Scientific). For each RT-qPCR reaction, 15 ng of cDNA was used with 0.1 µM of each primer and 10 µl of PowerUp™ SYBR™ Green Master Mix (Applied Biosystems) in a final volume of 20 µl. Samples were analysed using CFX Connect Real-Time PCR Detection System (Bio-Rad) using the following thermal cycling parameters: 3 min at 95°C, followed by 40 cycles of 10 s at 95°C, 20 s at 63°C followed by 30 s at 72°C. Each sample was run in triplicate. 18S rRNA was used as normaliser.

## Nuclear fractionation and co-immunoprecipitation

Cells were washed twice with ice-cold PBS, scraped and collected by centrifugation at 10,000 *g* for 1 min. Pellets were resuspended in 500 µl of Buffer A (10 mM HEPES pH 7.9, 1.5 mM $MgCl_2$, 10 mM KCl, 1 mM PMSF, 0.5 mM DTT, protease inhibitor cocktail) and cytoplasmic fractions were obtained by incubating the cells on ice for 15 min, adding of 0.05% NP-40, followed by centrifugation at 17,000 *g* for 10 min at 4°C. The resulting pellet was gently washed with Buffer A and resuspended in Buffer C (10 mM HEPES pH 7.9, 12.5% glycerol, 0.75 mM $MgCl_2$, 10 mM KCl, 0.1 mM EDTA, 400 mM NaCl, 1 mM PMSF, 0.5 mM DTT, protease inhibitor cocktail). Samples were incubated at 4°C with shaking at 3400 rpm

for 1 h, with vortexing every 10 min. Nuclear fractions were collected by centrifugation at 17,000 g for 10 min at 4°C.

For co-IP, 25 µl of Dynabeads™ Protein A (Thermo Fisher Scientific, #10002D) were washed twice with PBS containing 0.1% Tween-20 (PBS-T), resuspended in 1 ml of PBS-T, and incubated with 2 µg of the appropriate antibody (see Table S5) for 1 h at 4°C with rotation. Bead-antibody complexes were washed twice with PBS-T, resuspended in 125 µl PBS-T and incubated with 100 µg of nuclear extract for 3 h at 4°C with rotation. Beads were then washed five times with Buffer C and eluted in 25 µl of Buffer C supplemented with 4.1 µl of 6× reducing Laemmli SDS sample buffer (Thermo Fisher Scientific). Eluates and 5% input controls (nuclear extract diluted 1:10 in PBS and mixed with 1:6 volume of 6× Laemmli buffer) were boiled for 5 min before SDS-PAGE. For western blotting, 20 µl of each sample were loaded as described below.

## Western blot

For each condition, cells were harvested and lysed with cold radioimmunoprecipitation assay buffer (Pierce™ RIPA buffer, Thermo Fisher Scientific) with Pierce™ Protease Inhibitor Mini Tablets, EDTA-free (Thermo Fisher Scientific). After centrifugation at 4°C at 16,000 *g* for 10 min, the total extracted protein was quantified using Pierce™ BCA Protein Assay Kit, following the manufacturer's indications. We loaded 30 µg of total protein in Novex WedgeWell 4-20% Tris-Glycine Gel (Invitrogen) and separated it through gel electrophoresis (SDS–PAGE) Tris-Glycine-SDS buffer (Invitrogen). The proteins were then transferred to nitrocellulose membrane (Thermo Fisher Scientific) and incubated for 1 h in Intercept® (PBS) Blocking Buffer (Licor). After blocking, membranes were incubated overnight at 4°C (while rocking) in suitable antibody (see Table S5) diluted in blocking buffer and 0.2% Tween-20, at recommended concentrations. Membranes were washed in 0.1% PBS-T before and after being incubated for 1 h in the appropriate secondary antibody (1:15,000 dilution). The antibodies were then visualised using Odyssey XF Imaging System (Licor) or ChemiDocTM MP Imaging System and imaged with Image Studio Software (Licor) or Fiji (Schindelin et al., 2012).

## Flow cytometry analysis of surface markers

To obtain a single-cell suspension for flow cytometry analysis, cells were treated with Accutase for 5 min at 37°C. Cells were then washed with ice-cold PBS-2% foetal bovine serum (FBS) and live cells were counted. Then $1×10^6$ cells/condition were resuspended in 100 µl PBS-2% FBS and stained. For pluripotency evaluation, 4 µl of the respective antibodies were used. For analysis of differentiation, 2 µl of the respective antibodies were used. Cells were incubated for 15 min on ice and protected from light before being transferred into fluorescence-activated cell sorting (FACS) tubes containing an additional 300 µl PBS-2% FBS. Flow cytometry data were acquired using NovoCyte Penteon Flow Cytometer Systems 5 Lasers and analysed with FlowJo Software version 10.9. For antibodies and reagents see Table S5.

## Immunofluorescence

Cells were passaged onto Geltrex-coated coverslips and fixed at required confluence or time point. In detail, cells were incubated in 4% paraformaldehyde at 37°C for 15 min, followed by three 2-min washes in 1× PBS. For permeabilisation, cells were treated with 0.1% (v/v) Triton X-100 (T8787, Sigma-Aldrich) at room temperature for 10 min before three 2-min washes in PBS-T and a 1-h blocking upon 10% donkey serum (D9663, Sigma-Aldrich) at room temperature. Cells were incubated with corresponding PBS-diluted primary antibodies protected from direct light at 4°C overnight (see Table S5). After three 5-min washes with 1× PBS-T, cells were incubated for 1 h in PBS-diluted secondary antibodies and 15 min with DAPI (1:5000; 422801, BioLegend). Cells were then washed three times with 1× PBS-T for 5 min and once with 1× PBS. Coverslips were mounted onto slides using Fluorescence Mounting Medium (S3023, Agilent Technologies). Slides were imaged using confocal microscope (LSM780, Zeiss). The intensity of fluorescent signal was measured using corrected total cell fluorescence using Image J. Briefly, for each cell, a region of interest (ROI) was drawn around the cell boundary, and the integrated density (sum of all pixel intensity values within the ROI), area of ROI (total number of pixels contained in the ROI), and the mean

background fluorescence (average pixel intensity measured from a background region of the image lacking fluorescent signal) were measured.

The corrected fluorescence intensity was then calculated as:

$$CTCF = Integrated\ Density$$
$$- (Area\ of\ ROI\ \times\ Mean\ Background\ Fluorescence).$$

For colony-forming cells (e.g. iPSCs), we analysed segmented nuclei located within a section along the horizontal diameter of each colony, encompassing ~30-40 cells. This approach allowed us to capture differences between the colony centre and edges while enabling comparisons across colonies of similar size.

### RNA-seq
Total RNA was extracted using Monarch® Total RNA Miniprep Kit (New England Biolabs, T2010S) according to the manufacturer's instructions. RNA was quantified using NanoDrop 2000 UV Visible Spectrophotometer, while the RNA integrity was checked on Tapestation 2200 (Agilent). Only samples with an RNA integrity number (RIN) value above 8.0 were used for transcriptome analysis. RNA libraries were prepared using 1 μg of total RNA input using NEBNext® Poly(A) mRNA Magnetic Isolation Module, NEBNext® UltraTM II Directional RNA Library Prep Kit for Illumina® according to the manufacturer's instructions (New England Biolabs).

### RNA-sequencing analysis
After removing the adapters with TrimGalore! (https://www.bioinformatics.babraham.ac.uk/projects/trim_galore/), Kallisto was used to quantify reads mapping to each gene. Differential gene expression levels were determined using DESeq2 (Love et al., 2014). All statistical analyses were performed using the latest versions of BEDTools (Quinlan and Hall, 2010), deepTools (Ramírez et al., 2016), R Studio (version 4.4.3; Trophy Case) and GraphPad Prism (version 10.2.3; http://www.graphpad.com/faq/viewfaq.cfm?faq=1362).

### CUT&RUN
For each condition, $1\times10^5$ cells were processed using CUT&RUN Assay Kit (Cell Signaling Technology, #86652). Briefly, each sample was resuspended in Concanavalin A Beads and treated with an Antibody Binding Buffer and the suitable antibody (Table S5). After the antibody incubation, each reaction underwent a step of pAG-MNase enzyme binding, and subsequent DNA Digestion and diffusion. The diffused DNA was then purified using phenol/chloroform extraction and ethanol precipitation. The purified DNA was quantified using QuantiFluor dsDNA System, using the ONE dsDNA protocol (Promega). Libraries for sequencing were prepared using NEBNext Ultra II DNA Library Prep Kit for Illumina (New England Biolabs) and the libraries size distribution was assessed through automated electrophoresis using the Agilent 2200 Tapestation System (Agilent Technology).

### ChIP-seq
All samples from different conditions were processed together to prevent batch effects. Approximately 15 million cells were double cross-linked with 2 mM disuccinimidyl glutarate (DSG) for 20 min followed by 1% formaldehyde for 10 min at room temperature, quenched with 125 mM glycine, harvested, and washed twice with 1× PBS. The fixed cell pellet was resuspended in ChIP lysis buffer (150 mM NaCl, 1% Triton X-100, 0.7% SDS, 500 μM DTT, 10 mM Tris-HCl, 5 mM EDTA) and chromatin was sheared to an average length of 200-900 base-pairs, using a Covaris S220 Ultrasonicator at 5% duty factor for 7 min, three times. The chromatin lysate was diluted with SDS-free ChIP lysis buffer. For BRG1, 10 μg of antibody was used. The antibody was added to at least 5 μg of sonicated chromatin along with Dynabeads Protein G magnetic beads (Invitrogen) and incubated with rotation at 4°C overnight. The beads were washed twice with each of the following buffers: Mixed Micelle Buffer (150 mM NaCl, 1% Triton X-100, 0.2% SDS, 20 mM Tris-HCl, 5 mM EDTA, 65% sucrose), Buffer 200 (200 mM NaCl, 1% Triton X-100, 0.1% sodium deoxycholate, 25 mM HEPES, 10 mM Tris-HCl, 1 mM EDTA), LiCl detergent wash (250 mM LiCl, 0.5% sodium deoxycholate, 0.5% NP-40, 10 mM Tris-HCl, 1 mM

EDTA) and a final wash was performed with cold 0.1× Tris-EDTA buffer (TE). Finally, beads were resuspended in 1× TE containing 1% SDS and incubated at 65°C for 10 min to elute immunocomplexes. The elution was repeated twice, and the samples were incubated overnight at 65°C to reverse cross-linking, along with the input (5% of the starting material). The DNA was digested with 0.5 mg/ml Proteinase K for 3 h at 55°C and then purified using the ChIP DNA Clean & Concentrator kit (Zymo) and quantified with QUBIT. Barcoded libraries were made with NEBNext Ultra II DNA Library Prep Kit for Illumina using NEBNext Multiplex Oligos Dual Index Primers for Illumina (New England Biolabs) and sequenced on a NovaSeq X Plus instrument (Illumina).

### ATAC-seq
One hundred thousand cells/condition were harvested by enzymatic dissociation using StemPro™ Accutase™ (Thermo Fisher Scientific). Each cell pellet was processed using ATAC-Seq Kit (Active Motif), following the manufacturer's directions. Briefly, the fresh cell pellets were washed in 1× PBS and lysate in ATAC Lysis Buffer. Each lysate underwent a process of tagmentation and purification, before being amplified through PCR, using the following thermal cycler parameters: 72°C for 5 min, 98°C for 30 s, followed by 10 cycles of 98°C for 10 s, 63°C for 30 s, 72°C for 1 min. Each PCR reaction was cleaned up using SPRI beads (1.2× of the sample volume). The Agilent 2200 Tapestation System (Agilent Technologies) was used as quality control tool before performing next generation sequencing (NGS).

### Bioinformatics analysis of CHIP-seq, ATAC-seq and CUT&RUN data
Adapters were removed with TrimGalore! (https://www.bioinformatics.babraham.ac.uk/projects/trim_galore/), the sequences were aligned to the reference hg19, using Burrows-Wheeler Alignment tool, with the MEM algorithm (Herbreteau et al., 2021). Uniquely mapping aligned reads were filtered based on mapping quality (MAPQ>10) to restrict our analysis to higher quality and uniquely mapped reads, and PCR duplicates were removed. MACS2 (Zhang et al., 2008) was used to call peaks using the default parameters at 5% FDR.

### Motif analysis
BED files for the ROIs were produced using BEDTools (Quinlan and Hall, 2010). Motif analysis was performed using HOMER (Heinz et al., 2010). Shuffled input sequences were used as background. $E$-values<0.001 were used as a threshold for significance.

### Quantification and statistical analysis
RT-qPCR and image quantification data were analysed using GraphPad Prism software version 10.2.3 (http://www.graphpad.com/faq/viewfaq.cfm?faq=1362). Data are represented as mean±standard error of mean (s.e.m.). GO enrichment analysis was performed using Webgestalt 2019 (Liao et al., 2019). All statistical analyses were performed using the latest versions of BEDTools (Quinlan and Hall, 2010), Deeptools (Ramírez et al., 2016), R version 4.4.3 (Trophy Case; https://www.R-project.org/) and Prism software version 10.2.3. In particular, Bigwigs were generated with Deeptools and normalised by sequencing depth using the same software, and upper-quantile normalised using CHIPIN (Polit et al., 2021).

### Acknowledgements
We thank the Facility for Imaging by Light Microscopy (FILM) at Imperial College London for providing access to instrumentation and technical support, which is part-supported by funding from the Wellcome Trust (grant 104931/Z/14/Z) and Biotechnology and Biological Sciences Research Council (grant BB/L015129/1). We thank Prof. Brian Hendrich (University of Cambridge) and Dr Andria Koulle (Imperial College London) for insightful discussion of the data and of early versions of the manuscript.

### Competing interests
The authors declare no competing or financial interests.

## Author contributions
Conceptualization: M.D., M.T.; Data curation: M.D., M.T.; Formal analysis: M.D., M.T.; Funding acquisition: M.T.; Investigation: M.D., S.M.B., E.v.D., Z.H.M., L.D., M.T.; Methodology: M.D., M.T.; Project administration: M.T.; Resources: M.T.; Supervision: M.T.; Writing – original draft: M.D., M.T.; Writing – review & editing: S.M.B., E.v.D., Z.H.M., L.D.

## Funding
For this work, M.T. was funded by the G. Harold and Leila Y. Mathers Foundation and by the Biotechnology and Biological Sciences Research Council (grant BB/Y000854/1). Open Access funding provided by Imperial College London. Deposited in PMC for immediate release.

## Data and resource availability
RNA-seq, ATAC-seq, CUT&RUN-seq and ChIP-seq data have been deposited with GEO under accession code GSE293821. All other relevant data and details of resources can be found within the article and its supplementary information.

## The people behind the papers
This article has an associated 'The people behind the papers' interview with some of the authors.

## Peer review history
The peer review history is available online at https://journals.biologists.com/dev/lookup/doi/10.1242/dev.205248.reviewer-comments.pdf

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
