## [Peer Review File · Development (Cambridge, England)]

Neural crest induction requires SALL4-mediated BAF recruitment to lineage specific enhancers

Martina Demurtas, Samantha M. Barnada, Emma Van Domselaar, Zoe H. Mitchell, Laura Deelen and Marco Trizzino
DOI: 10.1242/dev.205248

Editor: Benoit G. Bruneau

Review timeline

Submission to Review Commons:	13 June 2025
Submission to Development:	16 September 2025
Editorial Decision:	10 October 2025
First revision received:	14 November 2025
Accepted:	14 November 2025

Reviewer 1

Evidence, reproducibility and clarity

Summary: The authors have previously published Mass-spectrometry data that demonstrates a physical interaction between Sall4 and the BAF chromatin complex in iPSC derived neurectodermal cells that are a precursor cell state to neural crest cells. The authors sought to understand the basis of this interaction and investigate the role of Sall4 and the BAF chromatin remodelling complex during neural crest cell specification.

The authors first validate this interaction with a co-IP between ARID1B subunit and Sall4 confirming the mass spec data. The authors then utilise in silico modelling to identify the specific interaction between the BAF complex and Sall4, suggesting that this contact is mediated through the BAF complex member DPF2.

To functionally validate the role of Sall4 during neural crest specification, the authors utilise CRISPR-Cas9 to introduce a premature stop codon on one allele of Sall4 to generate iPSCs that are haploinsufficient for Sall4.

Due to the reports of Sall4's role in pluripotency, the authors confirm that this model doesn't disrupt pluripotent stem cells and is viable to model the role of Sall4 during neural crest induction.

The authors expand this assessment of Sall4 function further during their differentiation model to cranial neural crest cells, assessing Sall4 binding with Cut+Run sequencing, revealing that Sall4 binds to motifs that correspond to key genes in neural crest differentiation. Moreover, reduction in Sall4 expression also reduces the binding of the BAF complex, through Cut and Run for BRG1. Overall, the authors then propose a model by which Sall4 and BRG1 bind to and open enhancer regions in neurectodermal cells that enable complete differentiation to cranial neural crest cells.

Overall, the data is clear and reproducible and offers a unique insight into the role of chromatin remodellers during cell fate specification.

However, I have some minor comments.

1. Using AlphaFold in silico modelling, the authors propose the interaction between the BAF complex with Sall4 is mediated by DPF2, but don't test it. Does a knockout, or knockdown of DPF2 prevent the interaction?
2. OPTIONAL: Does knockout of DPF2 phenocopy the Sall4 ko? This would be very interesting to

include in the manuscript, but it would perhaps be a larger body of work.

3. Figure 1, the day of IP is not clearly described until later in the text. please outline during in the figure 3- What is the expression of Sal1 (and other Sall paralogs) during differentiation. The same with the protein levels of Sall4, does this remain at the below 50%, or is this just during pluripotency?

4. The authors hypothesise that Sall4 binds to enhancers- with the criteria for an enhancer being that these peaks > 1KB from the TSS are enhancers. Can this be reinforced by overlaying with other ChIP tracks that would give more confidence in this? There are several datasets from Joanna Wysocka's lab that also utilise this protocol which can give you more evidence to reinforce the claim and provide further detail as to the role of Sall4

5. The authors state that cells fail to become cranial neural crest cells, however they do not propose what the cells do instead. do they become neural? Or they stay at pluripotency, which is one option given the higher expression of Nanog, OCT4 and OTX2 that are all expressed in pluripotent stem cells.

6. In general, I would like to see the gating strategy and controls for the flow cytometry in a supplemental figure.

7. For supplementary figure 1- please include the gene names in the main image panels rather than just the germ layer.

Significance

The strength of this study lies in its well-designed and clearly presented experiments and datasets. In particular, identifying the specific SALL4 isoform that interacts with the BAF complex- and further exploring the implications of this interaction-is a major highlight. The authors also make effective use of in silico modelling with AlphaFold, offering valuable mechanistic insight into how this interaction is mediated.

The topic should have appeal to researchers in developmental biology and epigenetics. This study represents a significant step forward in validating the interaction between SALL4 and the BAF complex, and it highlights the requirement of SALL4 for BAF-mediated chromatin remodelling during neural crest specification. These findings are likely to be of interest to those studying the gene regulatory mechanisms underlying craniofacial development.

However, while the authors outline the roles of SALL4 and the BAF complex in chromatin remodeling during neural crest development, the downstream effects on cell fate specification could be more thoroughly examined. Currently, Gene Ontology analysis is the primary method used to interpret these consequences, and additional functional validation would strengthen the conclusions.

Intended audience: Basic research, epigenetics in pluripotency and neural crest development.

Reviewer 2

Evidence, reproducibility and clarity

Summary

In this manuscript, the authors build on their previous work (Pagliaroli et al., 2021) where they identified an interaction between the transcription factor SALL4 and the BAF chromatin remodeling complex at Day-5 of an iPSC to CNCC differentiation protocol. In their current work, the authors begin by exploring this interaction further, leveraging AlphaFold to predict interaction surfaces between SALL4 and BAF complex members, considering both SALL4 splice isoforms: a longer SALL4A (associated with developmental processes) and a shorter SALL4B (associated with pluripotency). They propose that SALL4A may interact with DPF2, a BAF complex member, in an isoform-dependent manner. The authors next explore the role of SALL4 in craniofacial development, motivated by patient heterozygous loss of function mutations, leveraging iPSC cells with an engineered SALL4 frameshift mutation (SALL4-het- KO). Using this model, the authors first demonstrate that a reduced expression of SALL4 does not impact the iPSC identity, perhaps due to compensation via upregulation of SALL1.

Upon differentiation to neuroectoderm, SALL4 haploinsufficiency causes a reduction in newly accessible sites which are associated with a reduction in SALL4 binding and therefore a loss of BAF complex recruitment. Interestingly, however, there were few transcriptional changes at this stage. Later in the CNCC differentiation at Day-14 when the wildtype cells have switched expression of CNCC markers, the SALL4-het-KO cells fail to switch cadherin expression associated with a transition from epithelial to mesenchymal state, and fail to induce CNCC specification and post-migratory markers. Together the authors propose that SALL4 recruits BAF to CNCC enhancers as early as the neuroectodermal stage, and failure of BAF recruitment in SALL4-het-KO lines results in a loss of open chromatin at regulatory regions required later for induction of the CNCC programme. The failure of the later differentiation is compelling in the light of the early stages of the differentiation progressing normally, and the authors outline an interesting proposed mechanism whereby SALL4 recruits BAF to remodel chromatin ahead of CNCC enhancer activation, a model that can be tested further in future work.

Major comments

The link between SALL4 DNA binding and BAF recruitment is nicely argued, and very interesting as altered chromatin accessibility at Day 5 in the neuroectodermal stage is associated with only few changes in gene expression, while gene expression is greatly impacted later in the CNCC stage at Day 14. The *in silico* predictions of SALL4-BAF interaction interfaces are perhaps less convincing, requiring experimental follow-up outside the scope of this paper. Some of the associated figures could perhaps be moved to the supplement to enhance the focus on the later functional genomics experiments.

1. A lot of emphasis is placed on the AlphaFold predictions in Figure 1, however the predictions in Figure 1B appear to be mostly low or very low confidence scores (coloured yellow and orange). It is unclear how much weight can be placed on these predictions without functional follow-up, e.g. mutating certain residues and showing impact on the interaction by co-IP. The latter parts of the manuscript are much better supported experimentally, and therefore perhaps some of the Figure 1 could move to a Supplemental Figure (e.g. the right-hand part of 1B, and the lower part of Figure 1C showing SALL4B predicted interactions). The limitations of AlphaFold predictions should be acknowledged and the authors should discuss how these predicted interactions could be experimentally explored further in the future.
2. The authors only show data for one heterozygous knockout clone for SALL4. It is usual to have more than one clone to mitigate potential clonal effects. The authors should comment why they only have one clone and include any data for a second clone for key experiments if they already have this. Alternatively, the authors could provide any quality control information generated during production of this line, for example if any additional genotyping was performed.
3. The authors show all genomics data (ATAC-seq, CUT&RUN and ChIP-seq) as heatmaps and average profiles. It would be valuable to see some representative loci for the ATAC seq (perhaps along with SALL4 and BRG1 recruitment) at some representative and interesting loci.
4. Figure 4A. The schematic could be improved by including brightfield or immunofluorescent images at the three stages of the differentiation. Are the iPS cells seeded as single cells, or passaged as colonies before starting the differentiation. Further details are required in the methods to clarify how the differentiation is performed, for example at what Day are the differentiating cells passaged, this is not shown on the schematic in Figure 4A.
5. There is likely some heterogeneity of cell types in the differentiation at Day 5 and Day 14. Can the authors comment on this from previous publications or perhaps conduct some IF for markers to demonstrate what proportions of cells are neuroectoderm at Day 5 and CNCCs at Day 14.
6. For the motif analysis for Day 5-specific SALL4 binding sites (Figure 4E), was *de novo* motif calling performed? Were any binding sites reminiscent of a SALL4 binding site observed (e.g. an AT-rich motif)? Could the authors comment on this in the text - if there is no SALL4 binding motif, does this suggest SALL4 is recruited indirectly to these sites via interaction with another transcription factor for example?
7. Does SALL1 remain upregulated at Day-5 and Day-14 of the differentiation for the SALL4- het-KO line? Are binding sites known for this TF and were they detected in the motif analysis performed? Further discussion of the impact of the overexpression of SALL1 on the phenotypes observed is warranted - e.g. for Figure 5F, could the sites associated with a gain of BRG1 peaks upon loss of SALL4 be associated with SALL1 being upregulated and 'hijacking' BAF recruitment to distinct sites associated with nervous system development? Is SALL1 still upregulated at Day 5?

8. Related to the point above, SALL4A is proposed to have an isoform-specific interaction with the BAF complex. It would be valuable to plot SALL4A and SALL4B expression from the available RNA-seq data at Day 0, 5 and 14 to explore whether stage-specific isoform expression matches with the proposed role of SALL4A to interact with BAF at Day 5. It could be valuable to also look at expression of SALL1, 2 and 3 across the time course to see whether additional compensation mechanisms are at play during the differentiation.

9. At line 264, The authors state "SALL4 recruits the BAF complex at CNCC developmental enhancers to increase chromatin accessibility". Given that this analysis is performed at Day 5 of the differentiation, which is labelled as neuroectoderm what evidence do the authors have that these are specifically CNCC enhancers? Statements relating to enhancers should generally be re-phrased to putative enhancers (as no functional evidence is provided for enhancer activity), and further evidence could be provided to support that these are CNCC-specific regulatory elements, e.g. showing representative gene loci from CNCC-specific genes. Discussion of the RNA-seq presented in Supplementary Figure 2B may also be appropriate to introduce here given that large numbers of accessible chromatin sites are detected while the expression of very few genes is impacted, suggesting these sites may become active enhancers at a later developmental stage.

10. Do any of the putative CNCC enhancers detected at Day 5 as being sensitive to SALL4 downregulation and loss of BAF recruitment overlap with previously tested VISTA enhancers (<https://enhancer.lbl.gov/vista/>)?

Minor comments

1. The authors are missing references in the introduction "a subpopulation of neural crest cells that migrate dorsolaterally to give rise to the cartilage and bones of the face and anterior skull, as well as cranial neurons and glia".

2. The discussion of congenital malformations associated with SALL4 haploinsufficiency is brief in the introduction. From OMIM, SALL4 heterozygous mutations are implicated with the condition Duane-radial ray syndrome (DRRS) with "upper limb anomalies, ocular anomalies, and, in some cases, renal anomalies... The ocular anomalies usually include Duane anomaly". That Duane anomaly is one phenotype among a number for patients with SALL4 haploinsufficiency could be clarified in the introduction. Of note, this is stated more clearly in the discussion but needs re-wording in the introduction.

3. The statements "show that the SALL4A isoform directly interacts with the BAF complex subunit DPF2 through its zinc-finger-3 domain" and "this interaction occurs between the zinc-finger-cluster-3 (ZFC3) domain of SALL4A and the plant homeodomains (PHDs) of DPF2" in the introduction appear overstated and should be toned down. To show this the authors would need to mutate or delete the proposed important zinc-finger domains from SALL4A, which is outside the scope of this work. Notably, this is less strongly-stated elsewhere in the manuscript, e.g. "predict that this interaction is mediated by the BAF subunit DPF2", Line 162.

4. Could the authors clarify why 3 AlphaFold output models are shown for SALL4B in Figure 1C, and only one output model for SALL4A?

5. Line 121. The authors state "DPF2, a broadly expressed BAF subunit," but don't show expression during their CNCC differentiation. It would be good to include expression of DPF2 in Figure 1E.

6. The text states "a 11 bp deletion within the 3'-terminus of exon 1 of SALL4", while the figure legend states, "Sanger sequencing confirming the 19 bp deletion in one allele of SALL4 is displayed". The authors should clarify this disparity and experimentally confirm the deletion, e.g. by TA-cloning the two alleles and sequencing these separately to show that one allele is wildtype and the other has a frameshift deletion.

7. The authors generate an 11-bp (or 19-bp?) deletion in exon-1 - it would be valuable to include a discussion whether patients have been identified with deletions and frame-shift mutations in this region of SALL4 exon-1. And also clarify, if not clearly stated in the text, that both SALL4A and SALL4B will be impacted by this mutation. Are there examples of patient mutations which only impact SALL4A?

8. For the SALL4 blots in Figure 2B, is the antibody expected to detect both isoforms (SALL4A and SALL4B), and which isoform is shown? If two isoforms are detected, they should both be presented in the figure.

9. SALL4 expression should be shown for Figure 2C to see whether the >50% down-regulation of SALL4 at the protein level may be partially driven by transcriptional changes.

10. The number of experimental replicates should be indicated in all figure legends where relevant. Raw data points should be plotted visibly over the violin plots (e.g. Figure 2C).

11. For Figure 3A, the images of the DAPI and NANOG/OCT4 staining should be shown separately in addition to the overlay.
12. The metric 'Corrected Total Cell Fluorescence (CTCF)' should be described in the methods. The number of images used for the quantification in Figure 3A should be indicated in the legend, and error bars included if multiple images were quantified.
13. Figure 3C - what are the 114 differentially expressed genes? Some interesting genes could be labelled on the plot and the data used to generate this plot should be included as a Supplementary Table. Supplementary Tables should similarly be provided for Figure 6C, Day 14 and Supplementary Figure 2B, Day 5.
14. Figure 4B. The shared peaks are not shown. For completeness, it would be ideal to show these sites also.
15. Figure 4C is difficult to interpret. Why is the plot asymmetric to the left versus right? What does the axis represent - % of binding sites?
16. Line 224-225. What do $n=3,729$ and $n=6,860$ refer to? There appear to be many more binding sites indicated in Figure 4B, therefore these numbers cannot represent 86% and 97% of sites?
17. Figure 4E. Several TFs mentioned in the text (Line 243) are not shown in the figure, it would be good to show all TFs motifs mentioned in the text in this figure. Again, there is no mention of whether a sequence-specific motif is detected for SALL4 (e.g. an AT-rich sequence) from this motif analysis.
18. Figure 4G. How was the ATAC-seq data normalized for the WT and SALL4-het-KO lines for this comparison? The background levels of accessibility seem quite different in Replicate 1.
19. Figures 5B-C could be exchanged to flow better with the text. A Venn diagram could be included to show the overlap between the sites losing BRG1 in SALL4-het-KO (13,505 sites) and the Day5-specific SALL4 sites (17,137 sites).
20. At Day 5, the authors suggest a shift towards neural differentiation. It could be interesting for the authors to perform qRT-PCR at Day 5 for some neural markers or look in the Day 14 data for markers of neural differentiation at the expense of CNCC markers.
21. Is the data used to plot Figure 5D the same as Figure 4G. If so, why is only one replicate shown in Figure 5D?
22. Figure 6A. How many replicates are shown? If $n=2$, boxplots are not an appropriate to represent the distribution of the data. Please include $n=X$ in the figure legend and plot the raw data points also.
23. Figure 6B. What is the significance of CD99 for CNCC differentiation?
24. Figure 6F. No error bars are shown, how many replicates were performed for this time course? The linear regression line does not appear to add much value and could be removed.
25. At line 304, the authors state "while SALL4-het-KO showed a significant downregulation of these genes". Perhaps 'failed to induce these genes' may be more accurate unless they were expressed at Day 5 and downregulated at Day 14.
26. Lines 332-335. The genes selected for pluripotency, neural plate border, CNCC specification could be plotted separately in the Supplement to show individual gene expression dynamics.

Significance

This work provides a conceptual advance in understanding the aetiology of human SALL4-mediated craniofacial malformations in a cell-type specific manner. Leveraging an *in vitro* differentiation system, the authors define development timepoints and cell types impacted by altered SALL4 dosage. Additionally, the authors provide interesting mechanistic insights how the teratogen thalidomide may impact craniofacial development through proteasomal targeting and degradation of SALL4, and subsequent impact on neural crest differentiation progression.

Several audiences will be interested in this work: stem cell and developmental biologists (especially those interested in neural crest and facial development), and researchers interested in enhancer regulation, chromatin biology or gene regulatory mechanisms. Clinician scientists and geneticists will be interested in the proposed implications for mechanisms of disease.

Field of expertise: We have expertise in mechanisms of gene regulation and *in vitro* models of early development. We are not experts in modeling protein interactions *in silico*.

Author response to reviewers' comments

Reviewer #1

Summary: The authors have previously published Mass-spectrometry data that demonstrates a physical interaction between Sall4 and the BAF chromatin complex in iPSC derived neuroectodermal cells that are a precursor cell state to neural crest cells. The authors sought to understand the basis of this interaction and investigate the role of Sall4 and the BAF chromatin remodelling complex during neural crest cell specification.

The authors first validate this interaction with a co-IP between ARID1B subunit and Sall4 confirming the mass spec data. The authors then utilise in silico modelling to identify the specific interaction between the BAF complex and Sall4, suggesting that this contact is mediated through the BAF complex member DPF2.

To functionally validate the role of Sall4 during neural crest specification, the authors utilise CRISPR-Cas9 to introduce a premature stop codon on one allele of Sall4 to generate iPSCs that are haploinsufficient for Sall4.

Due to the reports of Sall4's role in pluripotency, the authors confirm that this model doesn't disrupt pluripotent stem cells and is viable to model the role of Sall4 during neural crest induction.

The authors expand this assessment of Sall4 function further during their differentiation model to cranial neural crest cells, assessing Sall4 binding with Cut+Run sequencing, revealing that Sall4 binds to motifs that correspond to key genes in neural crest differentiation. Moreover, reduction in Sall4 expression also reduces the binding of the BAF complex, through Cut and Run for BRG1.

Overall, the authors then propose a model by which Sall4 and BRG1 bind to and open enhancer regions in neuroectodermal cells that enable complete differentiation to cranial neural crest cells.

Overall, the data is clear and reproducible and offers a unique insight into the role of chromatin remodellers during cell fate specification.

We thank the Reviewer for the nice words of appreciation of our manuscript.

However, I have some minor comments.

1- Using AlphaFold in silico modelling, the authors propose the interaction between the BAF complex with Sall4 is mediated by DPF2, but don't test it. Does a knockout, or knockdown of DPF2 prevent the interaction?

We agree with the Reviewer that we are not functionally validating our computational prediction that DPF2 is the specific BAF subunit directly linking SALL4 with BAF. We chose not to perform the validation experiment for two main reasons:

1) This would be outside of the scope of the paper. In fact, from a mechanistic point of view, we have confirmed via both Mass-spectrometry and co-IP with ARID1B that SALL4 and BAF interact in our system. Moreover, mechanistically we also extensively demonstrate that the interaction with SALL4 is required to recruit BAF at the neural crest induction enhancers and we further demonstrate that depletion of SALL4 impairs this. In our view, this was the focus of the manuscript. On the other hand, detecting with certainty which BAF subunit mediates the interaction with SALL4 would be outside the scope of the paper.

2) Moreover, after careful consideration, we don't think that even a knock-out of DPF2 would provide a definite answer to which exact BAF subunit mediates the interaction with SALL4. In fact, knock out of DPF2 could potentially disrupt BAF assembly or stability, and this could result in a disruption of the interaction with SALL4 even if DPF2 is not the very subunit mediating it (in other words the experiment could provide a false positive result). In our opinion, the only effective experiment would be mutating the DPF2 residues that we computationally predicted as responsible for the interaction with SALL4, but again this would be very laborious and out of the scope.

That being said, we agree with the Reviewer that while the SALL4-BAF interaction was experimentally validated with robust approaches, the role of DPF2 in the interaction was only computationally predicted, which comes as a limitation of the study. We have now added a

dedicated paragraph in the discussion to acknowledge such limitation.

2- OPTIONAL: Does knockout of DPF2 phenocopy the Sall4 ko? This would be very interesting to include in the manuscript, but it would perhaps be a larger body of work.

See point-1.

3- Figure 1, the day of IP is not clearly described until later in the test. please outline during in the figure.

We thank the Reviewer for pointing this out. This has been fixed.

3- What is the expression of Sall1 (and other Sall paralogs) during differentiation. The same with the protein levels of Sall4, does this remain at the below 50%, or is this just during pluripotency?

As Recommend by the Reviewer, we have performed time-course WB of SALL1 and SALL4. These experiments revealed that SALL1 remains very lowly expressed in wild-type conditions across time points and all the way through differentiation until CNCC (See updated supplementary Fig. S9). This is consistent with previous studies that demonstrated that SALL4, but not SALL1, is required for early mammalian development (see for example Miller et al. 2016, Development, and Koulle et al. 2025, Biorxiv). We performed the same time-course WB for SALL4 which revealed that SALL4 expression progressively decreases after day-5 (as expected) and it's very low at CNCC stage (day-14), therefore we would expect the KO to remain at even lower level at this stage.

4- The authors hypothesise that Sall4 binds to enhancers- with the criteria for an enhancer being that these peaks > 1KB from the TSS are enhancers. Can this be reinforced by overlaying with other CHIP tracks that would give more confidence in this? There are several datasets from Joanna Wysocka's lab that also utilise this protocol which can give you more evidence to reinforce the claim and provide further detail as to the role of Sall4.

We thank the Reviewer for this great suggestion. As recommended, we have used publicly available CHIP-seq data generated by the Wysocka lab (H3K4me1, H3K4m3) and also generated new H3K27ac CHIP-seq data as well. These experiments and analyses confirmed that these regions are putative CNCC enhancers (and a minority of them putative promoters), decorated with H3K4me1 and with progressive increase in H3K27ac after CNCC induction (day-5). See new Supplementary Figure S6.

5- The authors state that cells fail to become cranial neural crest cells, however they do not propose what the cells do instead. do they become neural? Or they stay at pluriopotent, which is one option given the higher expression of Nanog, OCT4 and OTX2 that are all expressed in pluripotent stem cells.

We think that it is likely a mix of both. There is a mixed bag of expression of pluripotency markers, but also high expression of neuroectodermal markers. This suggests that most cells safely reach the neuroectodermal stage but fail to go beyond that, while some of the cells simply do not differentiate or regress back to pluripotency. We would rather refrain on overinterpreting what the KO-cells become, as it is likely an aberrant cell type, but following the Reviewer's indication we have added a paragraph in the discussion to speculate on this.

6- In general, I would like to see the gating strategy and controls for the flow cytometry in a supplemental figure.

As Recommended by the Reviewer, we have added the gating strategy in the Supplementary Fig. S4.

7- For supplementary figure 1- please include the gene names in the main image panels rather than just the germ layer.

Done. The figure is now Supplementary Figure S3 since two supplementary figures were added before.

Reviewer #2**Summary**

In this manuscript, the authors build on their previous work (Pagliaroli et al., 2021) where they identified an interaction between the transcription factor SALL4 and the BAF chromatin remodeling complex at Day-5 of an iPSC to CNCC differentiation protocol. In their current work, the authors begin by exploring this interaction further, leveraging AlphaFold to predict interaction surfaces between SALL4 and BAF complex members, considering both SALL4 splice isoforms: a longer SALL4A (associated with developmental processes) and a shorter SALL4B (associated with pluripotency). They propose that SALL4A may interact with DPF2, a BAF complex member, in an isoform-dependent manner. The authors next explore the role of SALL4 in craniofacial development, motivated by patient heterozygous loss of function mutations, leveraging iPSC cells with an engineered SALL4 frameshift mutation (SALL4-het-KO). Using this model, the authors first demonstrate that a reduced expression of SALL4 does not impact the iPSC identity, perhaps due to compensation via upregulation of SALL1. Upon differentiation to neuroectoderm, SALL4 haploinsufficiency causes a reduction in newly accessible sites which are associated with a reduction in SALL4 binding and therefore a loss of BAF complex recruitment. Interestingly, however, there were few transcriptional changes at this stage. Later in the CNCC differentiation at Day-14 when the wildtype cells have switched expression of CNCC markers, the SALL4-het-KO cells fail to switch cadherin expression associated with a transition from epithelial to mesenchymal state, and fail to induce CNCC specification and post-migratory markers. Together the authors propose that SALL4 recruits BAF to CNCC enhancers as early as the neuroectodermal stage, and failure of BAF recruitment in SALL4-het-KO lines results in a loss of open chromatin at regulatory regions required later for induction of the CNCC programme. The failure of the later differentiation is compelling in the light of the early stages of the differentiation progressing normally, and the authors outline an interesting proposed mechanism whereby SALL4 recruits BAF to remodel chromatin ahead of CNCC enhancer activation, a model that can be tested further in future work. The link between SALL4 DNA binding and BAF recruitment is nicely argued, and very interesting as altered chromatin accessibility at Day 5 in the neuroectodermal stage is associated with only few changes in gene expression, while gene expression is greatly impacted later in the CNCC stage at Day 14. The in silico predictions of SALL4-BAF interaction interfaces are perhaps less convincing, requiring experimental follow-up outside the scope of this paper. Some of the associated figures could perhaps be moved to the supplement to enhance the focus on the later functional genomics experiments.

We thank the Reviewer for the nice words of appreciation of our manuscript.

Major comments

1. A lot of emphasis is placed on the AlphaFold predictions in Figure 1, however the predictions in Figure 1B appear to be mostly low or very low confidence scores (coloured yellow and orange). It is unclear how much weight can be placed on these predictions without functional follow-up, e.g. mutating certain residues and showing impact on the interaction by co-IP. The latter parts of the manuscript are much better supported experimentally, and therefore perhaps some of the Figure 1 could move to a Supplemental Figure (e.g. the right-hand part of 1B, and the lower part of Figure 1C showing SALL4B predicted interactions). The limitations of AlphaFold predictions should be acknowledged and the authors should discuss how these predicted interactions could be experimentally explored further in the future.

As recommended by the Reviewer, we have moved part of the AlphaFold predictions to Supplementary Figure S1, and we added a paragraph in the discussion to acknowledge the limitations of AlphaFold.

2. The authors only show data for one heterozygous knockout clone for SALL4. It is usual to have more than one clone to mitigate potential clonal effects. The authors should comment why they only have one clone and include any data for a second clone for key experiments if they already have this. Alternatively, the authors could provide any quality control information generated during

production of this line, for example if any additional genotyping was performed.

We apologize for the confusion and for our lack of clarify on this. We have used two clones (one generated with a 11 bp deletion, one with a 19 bp deletion, both in exon-1, see also the point 6 of your minor points). The two clones were used as biological replicates, so for example the two ATAC-seq replicates performed in each time point were performed with the two different clones, and the three RNA-seq replicates were performed with two technical replicates of the clone with the 11bp deletion and one replicate with the clone with 19 bp deletion. We have clarified this in the methods section of the manuscript and added a Supplementary Figure (S2) showing the editing strategy for the two clones. Thank you for catching it.

3. The authors show all genomics data (ATAC-seq, CUT&RUN and ChIP-seq) as heatmaps and average profiles. It would be valuable to see some representative loci for the ATAC seq (perhaps along with SALL4 and BRG1 recruitment) at some representative and interesting loci.

As recommended by the Reviewer, we have added Genome Browser screenshots of representative loci in Fig. 6.

4. Figure 4A. The schematic could be improved by including brightfield or immunofluorescent images at the three stages of the differentiation. Are the iPSC cells seeded as single cells, or passaged as colonies before starting the differentiation. Further details are required in the methods to clarify how the differentiation is performed, for example at what Day are the differentiating cells passaged, this is not shown on the schematic in Figure 4A.

As recommended, we added IF images in the Fig. 4A schematic, and added more details in the methods.

5. There is likely some heterogeneity of cell types in the differentiation at Day 5 and Day 14. Can the authors comment on this from previous publications or perhaps conduct some IF for markers to demonstrate what proportions of cells are neuroectoderm at Day 5 and CNCCs at Day 14.

The differentiation starts with single cells that aggregate to form neuroectodermal clusters, as per original protocol. The CNCCs that we obtain with this protocol homogeneously express CNCC markers, as shown by IF of SOX9 in Fig. 4A. For the day-5, as recommended we have added IF for PAX6 also showing homogeneous expression (Fig. 4A).

6. For the motif analysis for Day 5-specific SALL4 binding sites (Figure 4E), was de novo motif calling performed? Were any binding sites reminiscent of a SALL4 binding site observed (e.g. an AT-rich motif)? Could the authors comment on this in the text - if there is no SALL4 binding motif, does this suggest SALL4 is recruited indirectly to these sites via interaction with another transcription factor for example?

Similar to SALL4, SALL1 also recognizes AT-rich motifs. However, while we found AT-rich motifs as enriched in our day-5 motif analysis (in the regions that gain SALL4 binding upon differentiation), the enrichment is not particularly strong, and several other motifs are significantly more enriched, suggesting that, like the Reviewer mentioned, SALL4 might be recruited indirectly at these sites by other factors. We have added a paragraph on this in the discussion.

7. Does SALL1 remain upregulated at Day-5 and Day-14 of the differentiation for the SALL4-het-KO line? Are binding sites known for this TF and were they detected in the motif analysis performed? Further discussion of the impact of the overexpression of SALL1 on the phenotypes observed is warranted - e.g. for Figure 5F, could the sites associated with a gain of BRG1 peaks upon loss of SALL4 be associated with SALL1 being upregulated and 'hijacking' BAF recruitment to distinct sites associated with nervous system development? Is SALL1 still upregulated at Day 5?

As mentioned above, SALL1 also recognizes AT-rich motifs but similar to SALL4 also binds unspecifically, likely in cooperation with other TFs. Like the Reviewer suggested, it is certainly possible that some of the sites associated with a gain of BRG1 peaks upon loss of SALL4 could be associated with SALL1 being upregulated and 'hijacking' BAF recruitment to distinct sites.

While this is speculative, we have added a paragraph on this in the discussion.

8. Related to the point above, SALL4A is proposed to have an isoform-specific interaction with the BAF complex. It would be valuable to plot SALL4A and SALL4B expression from the available RNA-seq data at Day 0, 5 and 14 to explore whether stage-specific isoform expression matches with the proposed role of SALL4A to interact with BAF at Day 5. It could be valuable to also look at expression of SALL1, 2 and 3 across the time course to see whether additional compensation mechanisms are at play during the differentiation.

Thanks for suggesting this. We performed a time course analysis of isoform specific gene expression, which showed that SALL4B expression remains low throughout differentiation, while SALL4A expression increases upon differentiation cues and it remains at high levels until the end. We have added this to supplementary Fig. S9. Moreover, we have performed an additional experiment, using pomalidomide, which is a thalidomide derivative that selectively degrades SALL4A but not SALL4B. Notably, SALL4A degradation recapitulated the main findings obtained with the CRISPR-KO of SALL4, further supporting that SALL4A is the isoform involved in CNCC induction (see new Fig. 8).

9. At line 264, The authors state "SALL4 recruits the BAF complex at CNCC developmental enhancers to increase chromatin accessibility". Given that this analysis is performed at Day 5 of the differentiation, which is labelled as neuroectoderm what evidence do the authors have that these are specifically CNCC enhancers? Statements relating to enhancers should generally be re-phrased to putative enhancers (as no functional evidence is provided for enhancer activity), and further evidence could be provided to support that these are CNCC-specific regulatory elements, e.g. showing representative gene loci from CNCC-specific genes. Discussion of the RNA-seq presented in Supplementary Figure 2B may also be appropriate to introduce here given that large numbers of accessible chromatin sites are detected while the expression of very few genes is impacted, suggesting these sites may become active enhancers at a later developmental stage.

As also recommended by the other Reviewer, to further characterize these sites, we have used publicly available histone modification CHIP-seq data (H3K4me1, H3K4me3) generated by the Wysocka lab (H3K4me1, H3K4m3) and also generated new H3K27ac CHIP-seq data as well. These experiments and analyses confirmed that these regions are putative CNCC enhancers (and a minority of them putative promoters), all decorated with H3K4me1, and all showing progressive increase in H3K27ac after CNCC induction (day-5). See new Supplementary Figure S6.

10. Do any of the putative CNCC enhancers detected at Day 5 as being sensitive to SALL4 downregulation and loss of BAF recruitment overlap with previously tested VISTA enhancers (<https://enhancer.lbl.gov/vista/>)?

Yes, we have found examples of overlap and have included two of them in the updated Figure 6 as Genome Browser screenshots.

Minor comments

1. The authors are missing references in the introduction "a subpopulation of neural crest cells that migrate dorsolaterally to give rise to the cartilage and bones of the face and anterior skull, as well as cranial neurons and glia".

Fixed, thank you.

2. The discussion of congenital malformations associated with SALL4 haploinsufficiency is brief in the introduction. From OMIM, SALL4 heterozygous mutations are implicated with the condition Duane-radial ray syndrome (DRRS) with "upper limb anomalies, ocular anomalies, and, in some cases, renal anomalies... The ocular anomalies usually include Duane anomaly". That Duane anomaly is one phenotype among a number for patients with SALL4 haploinsufficiency could be clarified in the introduction. Of note, this is stated more clearly in the discussion but needs re-wording in the introduction.

Done, thank you.

3. The statements "show that the SALL4A isoform directly interacts with the BAF complex subunit DPF2 through its zinc-finger-3 domain" and "this interaction occurs between the zinc-finger-cluster-3 (ZFC3) domain of SALL4A and the plant homeodomains (PHDs) of DPF2" in the introduction appear overstated and should be toned down. To show this the authors would need to mutate or delete the proposed important zinc-finger domains from SALL4A, which is outside the scope of this work. Notably, this is less strongly-stated elsewhere in the manuscript, e.g "predict that this interaction is mediated by the BAF subunit DPF2", Line 162.

Done, thank you.

4. Could the authors clarify why 3 AlphaFold output models are shown for SALL4B in Figure 1C, and only one output model for SALL4A?

AlphaFold3 produces five separate predicted models per protein combination (e.g., Model_1 ... Model_4), each derived from slightly different network parameters or initializations. The final output prioritizes the model with the highest confidence score. This multi-model strategy enables the identification of the most robust conformation while providing a measure of structural uncertainty (as per GitHub documentation for AlphaFold3). We have conducted the same analysis for SALL4A as we did for SALL4B. Specifically, SALL4A interacts with the AT-rich DNA in models 0, 1, and 2, therefore models 3 and 4 were excluded. When analysing models 1 and 2, we found a higher number of residues involved in the interaction (>800 instead of 396). Similarly to model 0, only the interactions between residues belonging to an annotated functional domain (ZFs and PHDs) were considered.

In Model 1: SALL4A and DPF2 interact mainly through ZF6 and 7, and not 5 as Model 0.

In Model 2: SALL4A and DPF2 interact mainly through ZF5 and 6, and not 7 as Models 0. In contrast, this model shows an interaction with ZF1 not shown in the other two models, but with a higher PAE (31 average compared to 25 to 27 average of the other two ZFs).

Therefore, we considered Model 0 as it is the model with higher confidence and representative of all significant models (includes ZF5, 6, and 7).

5. Line 121. The authors state "DPF2, a broadly expressed BAF subunit," but don't show expression during their CNCC differentiation. It would be good to include expression of DPF2 in Figure 1E.

Done, thank you.

6. The text states "a 11 bp deletion within the 3'-terminus of exon 1 of SALL4", while the figure legend states, "Sanger sequencing confirming the 19 bp deletion in one allele of SALL4 is displayed". The authors should clarify this disparity and experimentally confirm the deletion, e.g. by TA-cloning the two alleles and sequencing these separately to show that one allele is wildtype and the other has a frameshift deletion.

We apologize for the confusion. As stated above (point-2 of the major comments), we have used two clones (one generated with a 11 bp deletion, one with a 19 bp deletion, both in exon-1, see also the point 6 of your minor points). The two clones were used as biological replicates (see response above for details). The deletion for both clones was experimentally confirmed by Sanger sequencing by the company that generated the lines for us (Synthego). The strategy for the two clones is now shown also in Supplementary Fig. S2.

7. The authors generate an 11-bp (or 19-bp?) deletion in exon-1 - it would be valuable to include a discussion whether patients have been identified with deletions and frame-shift mutations in this region of SALL4 exon-1. And also clarify, if not clearly stated in the text, that both SALL4A and SALL4B will be impacted by this mutation. Are there examples of patient mutations which only impact SALL4A?

As requested, we have added a discussion paragraph to discuss this. And, yes, both SALL4A and SALL4B are impacted by both deletions in both clones (11 bp and 19 bp deletion).

Regarding patient variants on exon-1 and patient variants that only impact SALL4A. We could only find one published pathogenic 170bp deletion in exon 1 (VCV000642045.7). The majority of the

pathogenic or likely pathogenic variances are located on exon2. In particular, of the 63 reported pathogenic (or likely pathogenic) clinical variants, 42 were located on exon 2. Among these, 28 are located in the portion shared by both SALL4A and SALL4B, while the remaining 14 were SALL4A specific.

8. For the SALL4 blots in Figure 2B, is the antibody expected to detect both isoforms (SALL4A and SALL4B), and which isoform is shown? If two isoforms are detected, they should both be presented in the figure.

Yes, the antibody detects both isoforms, and we now present both in the figure 2, as recommended.

9. SALL4 expression should be shown for Figure 2C to see whether the >50% down-regulation of SALL4 at the protein level may be partially driven by transcriptional changes.

Done, thank you. As expected, we observed the SALL4 mRNA expression in the KO line is comparable to wild-type conditions, but still this results in a significant decrease of the SALL4 protein level likely because of autoregulatory mechanisms coupled with non-sense mediated decay of the mutated allele. Also, we note that SALL4 usually makes homodimers, therefore lack of sufficient amount of protein could also lead to degradation of the monomers.

10. The number of experimental replicates should be indicated in all figure legends where relevant. Raw data points should be plotted visibly over the violin plots (e.g. Figure 2C).

Done, thank you.

11. For Figure 3A, the images of the DAPI and NANOG/OCT4 staining should be shown separately in addition to the overlay.

Done, thank you.

12. The metric 'Corrected Total Cell Fluorescence (CTCF)' should be described in the methods. The number of images used for the quantification in Figure 3A should be

Done, thank you.

13. Figure 3C - what are the 114 differentially expressed genes? Some interesting genes could be labelled on the plot and the data used to generate this plot should be included as a Supplementary Table. Supplementary Tables should similarly be provided for Figure 6C, Day 14 and Supplementary Figure 2B, Day 5.

As recommended, we have highlighted some interesting genes in the volcano plot and also included all the expression data for all genes in Supplementary Table S3.

14. Figure 4B. The shared peaks are not shown. For completeness, it would be ideal to show these sites also.

Done, thank you.

15. Figure 4C is difficult to interpret. Why is the plot asymmetric to the left versus right? What does the axis represent - % of binding sites?

The asymmetry is due to the fact that there is a larger number of peaks that are downstream of the TSS than peaks that are upstream of TSS. This is consistent with the fact that many SALL4 peaks are in introns, likely representing intronic enhancers.

16. Line 224-225. What do n= 3,729 and n= 6,860 refer to? There appear to be many more binding sites indicated in Figure 4B, therefore these numbers cannot represent 86% and 97% of sites?

Thank you for pointing this out, we should have specified in the text. Those numbers refer to the genes whose TSS is closest to each SALL4 peak. Notably, multiple peaks can share the same closest

TSS, hence the discrepancy between # of peaks and # of nearest genes.

Raw numbers:

- Day-0 RAW = 6,104 (peaks = 6,114);
- Day-5 RAW = 17,131 (peaks = 17,137).

Now raw data reported in Supplementary Table 4.

17. Figure 4E. Several TFs mentioned in the text (Line 243) are not shown in the figure, it would be good to show all TFs motifs mentioned in the text in this figure. Again, there is no mention of whether a sequence-specific motif is detected for SALL4 (e.g. an AT-rich sequence) from this motif analysis.

Done, thank you. An AT-rich sequence, resembling the SALL4 motif, was detected in a small minority of sites (this is now shown in Supplementary Figure S5), suggesting that SALL4 engages chromatin in a broad manner, going beyond its preferred motif, possibly in cooperation with other TFs. This is consistent with many studies that in mESCs have shown that SALL4 binds at OCT4/NANOG/SOX2 target motifs. This is now discussed in a dedicated paragraph in the discussion.

18. Figure 4G. How was the ATAC-seq data normalized for the WT and SALL4-het-KO lines for this comparison? The background levels of accessibility seem quite different in Replicate 1.

The bigwigs used to make the heatmaps are normalized by sequencing depth using the Deeptools Suite (normalization by RPKM).

19. Figures 5B-C could be exchanged to flow better with the text. A Venn diagram could be included to show the overlap between the sites losing BRG1 in SALL4-het-KO (13,505 sites) and the Day5-specific SALL4 sites (17,137 sites).

Done, thank you.

20. At Day 5, the authors suggest a shift towards neural differentiation. It could be interesting for the authors to perform qRT-PCR at Day 5 for some neural markers or look in the Day 14 data for markers of neural differentiation at the expense of CNCC markers.

See updated Supplementary Fig. S8, where we show timecourse expression of several genes, including neural markers.

21. Is the data used to plot Figure 5D the same as Figure 4G. If so, why is only one replicate shown in Figure 5D?

Only one replicate was shown in the main figure purely for lack of space, but the experiment was replicated twice (with the two different clones), and the results were exactly the same. See plots below for your convenience:

ATAC-seq at sites losing BRG1 in SALL4-het-KO

22. Figure 6A. How many replicates are shown? If $n=2$, boxplots are not an appropriate to represent the distribution of the data. Please include $n= X$ in the figure legend and plot the raw data points also.

Done, thank you, and as suggested we are no longer using boxplots for this panel.

23. Figure 6B. What is the significance of CD99 for CNCC differentiation?

24. Figure 6F. No error bars are shown, how many replicates were performed for this time course? The linear regression line does not appear to add much value and could be removed.

As suggested, we have removed these plots and replaced them with individual genes plots, which include error bars. See updated Supplementary Figure S8.

25. At line 304, the authors state "while SALL4-het-KO showed a significant downregulation of these genes". Perhaps 'failed to induce these genes' may be more accurate unless they were expressed at Day 5 and downregulated at Day 14.

Done, thank you.

26. Lines 332-335. The genes selected for pluripotency, neural plate border, CNCC specification could be plotted separately in the Supplement to show individual gene expression dynamics.

Done, thank you, see point 24.

Original submission

First decision letter

MS ID#: dev.205248

MS Title: Neural crest induction requires SALL4-mediated BAF recruitment to lineage specific enhancers

Authors: Martina Demurtas; Samantha M. Barnada; Emma Van Domselaar; Zoe H. Mitchell; Laura Deelen; Marco Trizzino

Dear Dr Trizzino,

Thank you for sending your manuscript to Development through Review Commons.

I have now received all the referees reports on the above manuscript, and have reached a decision. The referees' comments are appended below, or you can access them online: please go to

The overall evaluation is positive and we would like to publish a revised manuscript in Development, provided that the referees' comments can be satisfactorily addressed. Please attend to all of the reviewers' comments in your revised manuscript and detail them in your point-by-point response. I will not be sending the ms back to reviewers, but they make good points that should be addressed. If you do not agree with any of their criticisms or suggestions explain clearly why this is so. If it would be helpful, you are welcome to contact us to discuss your revision in greater detail. Please send us a point-by-point response indicating your plans for addressing the referees' comments, and we will look over this and provide further guidance.

Reviewer 1: The authors present a study looking at the role of Sall4 during neural crest specification. They combine in silico modelling, genome editing, iPSC based model of neural crest

specification and genomics to dissect the mechanism by which Sall4 acts to mediate neural crest cell differentiation. The authors find that Sall4 interacts with the BAF complex enabling chromatin remodelling to open neural crest enhancers enabling differentiation to proceed. Moreover, finding that the Sall4a isoform is critical for this and Sall4b is unable to compensate, consistent with previous reports.

Following the review commons process, the authors have clearly and effectively addressed all of the reviewers' comments in full. The manuscript reads very well and provides a very clear, and well constructed manuscript looking into the process by which chromatin remodellers act to specify cell fate decisions.

The data is clearly presented and is robust. The manuscript is clearly written with thoughtful conclusions only stated based on what the data do show and clearly highlighting where further work is required to address any gaps.

Following reading of both the review commons report, the authors response and this manuscript, I am happy that all comments have been addressed and incorporated into the manuscript.

Reviewer 2: We thank the authors for their point-by-point response to our comments. We are satisfied with many of the responses but have several remaining queries and suggested changes. We would like to see a further revised manuscript as Figure 1 is currently missing and we would wish to see a response to our comments below.

General comments:

* Please could the authors provide a version of the manuscript with all altered text highlighted in red, or a word document with tracked changes. It is currently quite challenging to identify the altered text in the revised manuscript.

* Figure 1 appears to be missing from the submitted version. Adding Figure numbers to the main figures would also aid navigation through the figures.

* We would like to request access to the new Supplementary Tables.

Point-by-point:

Major Point 1.

The updated Figure 1 is missing in the updated manuscript.

Major Point 2.

Thank you for clarifying that there are two replicate clones with distinct deletions. Is Clone F1 named as KO1, and Clone C4 named as KO2, or the reverse?

It would be valuable to include the clone number where only one clone is used, for example in the legend for Figure 2B; 2F; Figure 3A; Figure 6C/E/F; Figure 7B; Supplementary Fig 2B-D; Supplementary Fig 3C.

For Figure 2B, only one clone is shown for the knockdown of SALL4A expression, is there a blot for the other clone to demonstrate the knockdown?

Major Point 4.

Regarding the differentiation protocol, the methods section is still quite brief.

"dissociated and treated with CNCC differentiation media" - how were the cells dissociated, i.e. with which enzyme? Are the cells dissociated as clumps, or as single cells - are they then grown in suspension, or plated? Is the plate coated?

"Around day 7-8 of differentiations, cells were transitioned to CNCC early maintenance media" - are the cells adherent at this stage? Are both neuroectoderm and CNCCs present on the dish?

"Differentiating cells were passaged -1:6 with 0.5mM EDTA or Accutase Cell Detachment Solution" - which day were the cells passaged? Were both the neuroectoderm and CNCCs passaged?

An IF of PAX6 has been included in the updated Figure 4A, however no mention of why this marker was chosen is referenced in the text, and the antibody does not appear to be mentioned in the Methods.

Major Point 5.

In their response, the authors state "The differentiation starts with single cells that aggregate to form neuroectodermal clusters" - this is not explained currently in the methods, and should be outlined how the cells are made to be single cells and how the neuroectodermal clusters form. This appears to deviate from the cited protocol and so warrants more detail.

Major Point 7.

Can the authors discuss the discrepancy between the RNA-seq in Figure 2E and Western blot in Figure 2F that SALL1 appears highly upregulated at the protein level, but the fold-change is much smaller transcriptionally (even if the difference is significant).

Major Point 10.

The putative enhancer region shown at the PAX7 gene in Figure 6F appears to overlap with a predicted enhancer from ENCODE data rather than from the VISTA enhancer browser. The element hs1004 however was tested in the VISTA enhancer browser and exhibits craniofacial signal in the mouse. All VISTA enhancer locations can be downloaded from UCSC using the Table Browser (Mammal - Human - hg38 - Regulation - VISTA enhancers - vistaEnhancerBb). The authors could therefore check further if any more of their putative enhancers have been tested in the VISTA browser.

Minor point 5.

Figure 1 appears to be missing, therefore we cannot see the new plot of DPF2 expression.

Minor point 7.

As stated above, it would be useful to have a version of the updated manuscript highlighting the text changes to find the authors' edits.

Minor point 11.

Could a statistical test be applied to the Figure 3A quantification.

Minor point 12.

The authors have indicated n=30 in the figure legend, but this is unclear if this refers to 30 nuclei/cells or 30 images. If this is 30 nuclei, how were these 30 nuclei selected from the 1 image. It would be ideal to repeat this analysis for at least one additional image. If this analysis is from one image, the segmented nuclei could be shown in the Supplementary Figure.

Minor point 20.

In the new Supplementary Figure 8, the blue and purple lines are not identified as wildtype or mutant. This should be indicated in the figure, or written in the figure legend.

The sentence "BRG1 fails to localize to these sites and is instead partially redirected to neurodevelopmental enhancers, leading to a misallocation of chromatin remodeling activity, potentially leading to a shift towards default neural differentiation at the expense of CNCC induction and specification" can be explored in Supplementary Figure 8 through the inclusion of a set of neural marker genes. It would also be useful to indicate significant expression changes at the 3 timepoints for each displayed gene.

Minor point 23.

There was no answer provided to this comment.

Author response to reviewers' comments**Comments from the Reviewers:****Reviewer 1:**

The authors present a study looking at the role of Sall4 during neural crest specification. They combine in silico modelling, genome editing, iPSC-based model of neural crest specification and genomics to dissect the mechanism by which Sall4 acts to mediate neural crest cell differentiation. The authors find that Sall4 interacts with the BAF complex enabling chromatin remodelling to open neural crest enhancers enabling differentiation to proceed. Moreover, finding that the Sall4a isoform is critical for this and Sall4b is unable to compensate, consistent with previous reports. Following the review commons process, the authors have clearly and effectively addressed all of the reviewers' comments in full. The manuscript reads very well and provides a very clear, and well-constructed manuscript looking into the process by which chromatin remodellers act to specify cell fate decisions.

The data is clearly presented and is robust. The manuscript is clearly written with thoughtful conclusions only stated based on what the data do show and clearly highlighting where further work is required to address any gaps. Following reading of both the review commons report, the authors response and this manuscript, I am happy that all comments have been addressed and incorporated into the manuscript.

We thank the Reviewer for their kind comments and for approving the publication of our manuscript.

Reviewer 2:

We thank the authors for their point-by-point response to our comments. We are satisfied with many of the responses but have several remaining queries and suggested changes. We would like to see a further revised manuscript as Figure 1 is currently missing and we would wish to see a response to our comments below.

Major Point 1.

The updated Figure 1 is missing in the updated manuscript.

We are unsure how this happened, and we apologise, we do see Figure-1 in the submitted finalized PDF version of the previous submission. The Figure 1 is attached to this new resubmission as well.

Major Point 2.

Thank you for clarifying that there are two replicate clones with distinct deletions. Is Clone F1 named as KO1, and Clone C4 named as KO2, or the reverse?

Clone C4 is KO1 and Clone F1 is KO2. This is now specified in the manuscript (see methods).

It would be valuable to include the clone number where only one clone is used, for example in the legend for Figure 2B; 2F; Figure 3A; Figure 6C/E/F; Figure 7B; Supplementary Fig 2B-D; Supplementary Fig 3C.

Done.

For Figure 2B, only one clone is shown for the knockdown of SALL4A expression, is there a blot for the other clone to demonstrate the knockdown?

Yes, the blot was performed also for the other clone, and you can see it here, showing a significant decrease in SALL4 expression upon heterozygous KO of SALL4.

Major Point 4.

Regarding the differentiation protocol, the methods section is still quite brief. "dissociated and treated with CNCC differentiation media" - how were the cells dissociated, i.e. with which enzyme? Are the cells dissociated as clumps, or as single cells - are they then grown in suspension, or plated? Is the plate coated?

We apologise for the lack of details. We have now added further details to the methods section. The cells are dissociated as clumps as in the original protocol, grown in suspension and then they are moved to coated plates (geltrex) where the spheres attach and the neural crest cells migrate out of the spheres.

"Around day 7-8 of differentiations, cells were transitioned to CNCC early maintenance media" - are the cells adherent at this stage? Are both neuroectoderm and CNCCs present on the dish?

"Differentiating cells were passaged 1:6 with 0.5mM EDTA or Accutase Cell Detachment Solution" - which day were the cells passaged? Were both the neuroectoderm and CNCCs passaged?

As mentioned above, We have now added further details to the methods section. Yes, as in the original protocol from the Wysocka lab, the spheres attach around day 7-9. The neuroectodermal cells are located in the centre of the sphere, while the neural crest cells migrate out of the sphere.

An IF of PAX6 has been included in the updated Figure 4A, however no mention of why this marker was chosen is referenced in the text, and the antibody does not appear to be mentioned in the Methods.

PAX6 is expressed in the neuroectoderm and was thus chosen as marker of neuroectodermal spheres.

Major Point 5.

In their response, the authors state "The differentiation starts with single cells that aggregate to form neuroectodermal clusters" - this is not explained currently in the methods, and should be outlined how the cells are made to be single cells and how the neuroectodermal clusters form. This appears to deviate from the cited protocol and so warrants more detail.

Again, apologies for lack of clarity, the used protocol is now described in more detail in the methods.

Major Point 7.

Can the authors discuss the discrepancy between the RNA-seq in Figure 2E and Western blot in Figure 2F that SALL1 appears highly upregulated at the protein level, but the fold-change is much smaller transcriptionally (even if the difference is significant).

Thanks for this comment. This might be due post-transcriptional regulation. We have added a sentence on this in the manuscript.

Major Point 10.

The putative enhancer region shown at the PAX7 gene in Figure 6F appears to overlap with a predicted enhancer from ENCODE data rather than from the VISTA enhancer browser. The element hs1004 however was tested in the VISTA enhancer browser and exhibits craniofacial signal in the mouse. All VISTA enhancer locations can be downloaded from UCSC using the Table Browser (Mammal - Human - hg38 - Regu Regulation - VISTA enhancers - vistaEnhancerBb). The authors could therefore check further if any more of their putative enhancers have been tested in the VISTA browser.

Thanks for the suggestion. We believe that, in the interest of space, the number of shown examples is sufficient.

Minor point 5.

Figure 1 appears to be missing, therefore we cannot see the new plot of DPF2 expression.
Figure 1 was added.

Minor point 11.

Could a statistical test be applied to the Figure 3A quantification.

Done.

Minor point 12.

The authors have indicated n=30 in the figure legend, but this is unclear if this refers to 30 nuclei/cells or 30 images. If this is 30 nuclei, how were these 30 nuclei selected from the 1 image. It would be ideal to repeat this analysis for at least one additional image. If this analysis is from one image, the segmented nuclei could be shown in the Supplementary Figure.

For colony-forming cells (e.g., iPSCs), we analysed segmented nuclei located within a section along the horizontal diameter of each colony, encompassing approximately 30-40 cells. This approach allowed us to capture differences between the colony centre and edges while enabling comparisons across colonies of similar size (see methods).

Minor point 20.

In the new Supplementary Figure 8, the blue and purple lines are not identified as wildtype or mutant. This should be indicated in the figure, or written in the figure legend.

Done.

The sentence "BRG1 fails to localize to these sites and is instead partially redirected to neurodevelopmental enhancers, leading to a misallocation of chromatin remodeling activity, potentially leading to a shift towards default neural differentiation at the expense of CNCC induction and specification" can be explored in Supplementary Figure 8 through the inclusion of a set of neural marker genes. It would also be useful to indicate significant expression changes at the 3 timepoints for each displayed gene.

Done.

Minor point 23.

There was no answer provided to this comment.

Apologies for missing this point in the previous rebuttal. CD99 was examined because it is one of the markers recommended for iPSC-to-CNCC differentiation in the original paper by the Wysocka lab where the protocol was published (Prescott et al. 2015, Cell).

Second decision letter

MS ID#: dev.205248R1

MS Title: Neural crest induction requires SALL4-mediated BAF recruitment to lineage specific enhancers

Authors: Martina Demurtas; Samantha M. Barnada; Emma Van Domselaar; Zoe H. Mitchell; Laura Deelen; Marco Trizzino

ARTICLE TYPE: Review Commons Transfer

Dear Dr Trizzino,

I am happy to tell you that your manuscript has been accepted for publication in Development, pending our standard publication integrity checks.